# Compression-induced NF-κB activation sustains tumor cell survival in confinement by detoxifying aldehydes and promotes metastasis

Bing Liu[1,8], Min Liu[1,8], Yajuan Zhang [2], Yifei Zhu[3], Dingpei Zhou[1], Hong Gao[1], Fan Yang[4], Dong Gao [1], Yun Zhao [1], BangBao Tao [5] ✉, Feng Yao [2] ✉ & Weiwei Yang [1,6,7] ✉

Metastasis remains the primary cause of cancer-related mortality. During dissemination, cancer cells must navigate spatially confined microenvironments, yet the underlying metabolic adaptations that facilitate this process remain unclear. Here, through an in vivo CRISPR screen targeting metabolic enzymes, we identify aldehyde dehydrogenase 1 family member B1 (ALDH1B1) as essential for tumor cell survival in confining capillaries. Mechanistically, compressive force induces casein kinase 2 alpha 3 (CSK23) to phosphorylate kappa-B kinase subunit beta (IKKβ) at Ser177/181, which activates the nuclear factor kappa B (NF-κB) pathway and upregulates ALDH1B1. The upregulation of ALDH1B1 enhances aldehyde detoxification, which suppresses ferroptosis and promotes tumor cell survival during migration through the capillaries, thereby facilitating metastasis. Importantly, genetic or pharmacological inhibition of CSK23 or ALDH1B1 effectively impairs metastasis. In lung cancer patients, confined tumor cells exhibit higher levels of ALDH1B1 and NF-κB activation, which correlates with metastatic recurrence. Our findings reveal a mechano-metabolic pathway that promotes metastasis and suggest CSK23 and ALDH1B1 as potential therapeutic targets.

Tumor metastasis, the movement of tumor cells from a primary site to colonize distant organs, is a significant factor in the mortality of cancer patients[1]. The metastatic cascade is an intricate multi-step journey that requires cancer cells to first dissociate from the primary tumor, intravasate into circulation, survive transit, extravasate at distant sites, and ultimately colonize secondary tissues[2]. A pivotal component of this process is cell migration, which requires navigating spatially complex microenvironments[3]. The capacity of cancer cells to migrate within physically restricted compartments represents a critical mode of in vivo movement, such as through pores within the tumor extra-cellular matrix (ECM) or pre-existing tunnel-like tracks, including paths along interstitial ECM fibers, the spaces between muscle and nerve

[1]Key Laboratory of Multi-Cell Systems, Shanghai Key Laboratory of Molecular Andrology, Center for Excellence in Molecular Cell Science, Chinese Academy of Sciences, Shanghai Institute of Biochemistry and Cell Biology, Shanghai, China. [2]Department of Thoracic Surgery, Shanghai Chest Hospital, Shanghai Jiao Tong University, Shanghai, China. [3]Department of Oncology, Fudan University Shanghai Cancer Center, Shanghai, China. [4]Shenzhen Center for Disease Control and Prevention, Shenzhen, China. [5]Department of Neurosurgery, XinHua Hospital School of Medicine, Shanghai Jiaotong University, Shanghai, China. [6]Key Laboratory of Systems Health Science of Zhejiang Province, School of Life Science, Hangzhou Institute for Advanced Study, University of Chinese Academy of Sciences, Hangzhou, China. [7]Shanghai Academy of Natural Sciences (SANS), Shanghai, China. [8]These authors contributed equally: Bing Liu, Min Liu. ✉e-mail: taobangbao@xinhuamed.com.cn; yaofeng@shsmu.edu.cn; wyang@sibcb.ac.cn

fibers, routes adjacent to or within blood vessels, and the vascular networks of distant organs[4–8]. Consequently, a detailed understanding of the molecular machinery controlling this confined migration is vital for designing therapeutic approaches to block metastatic dissemination.

Cancer cells frequently alter their metabolic processes to sustain cell growth and viability[9]. Emerging evidence shows that metastasizing cancer cells adapt their metabolism selectively and dynamically at each step of the metastatic process[10]. For instance, the expression of glutamate dehydrogenase 1 (GDH1) is upregulated in circulating tumor cells (CTCs), which contributes to anti-anoikis signals and promotes metastatic behavior in liver kinase B1 (LKB1)-deficient lung cancer[11]. During the colonization of distant organs, breast cancer cells induce the production of α-ketoglutarate-activated collagen hydroxylation to drive collagen-based ECM remodeling in the lung metastatic niche[12]. We recently demonstrate that UDP-glucose 6-dehydrogenase (UGDH) enhances the stability of snail family transcriptional repressor 1 (SNAI1) mRNA to initiate the epithelial-mesenchymal transition, thus promoting the extravasation process of lung cancer cells. In contrast, it is mostly unknown how tumor cells rewire metabolism to support their migration through the spatially-confining environment, thereby accomplishing their seeding to distant organ.

In this study, we performed the in vivo CRISPR loss-of-function screening in lung cancer cells with sgRNA targeting 1685 metabolic enzymes and identified that aldehyde dehydrogenase 1 family member B1 (ALDH1B1) is required for tumor cell survival in confining capillaries. ALDH1B1 is an aldehyde dehydrogenase that detoxifies aldehydes into corresponding carboxylic acids in cells[13]. We demonstrate that ALDH1B1 expression enhances aldehydes detoxification and suppresses ferroptosis in confined cells, which supports tumor cell survival during their migration in confining capillaries, thereby promoting lung cancer metastasis. In response to compressive force, casein kinase 2 alpha 3 (CSK23) activates nuclear factor kappa B (NF-κB) pathway to upregulate ALDH1B1 expression.

## Results

### ALDH1B1 is essential for tumor cell survival in capillaries and subsequent distant metastasis

To identify the metabolic enzyme that is required for tumor cell migration in confining spaces in vivo, we conducted CRISPR knockout screening using lung adenocarcinoma-derived A549 cells-an epithelial cell line with robust migratory and invasive capacities[14,15]. Cells were transduced with a gRNA library targeting 1685 metabolic enzymes (Supplementary Data 1) with a lung metastasis mouse model via tail vein injection. 30 min after injection, almost all tumor cells have reached the lung tissues, 24 h after implantation, over 90% of tumor cells remained within the capillaries of lung tissues and approximately 20% of these cells survived in lung capillaries and were responsible for subsequent extravasation and distant metastasis[16–20]. GFP-expressing living tumor cells were then isolated from dissected lung tissues and gRNA abundance was quantified by the next-generation sequencing (NGS) (Fig. 1a, Supplementary Fig. 1a, b, and Supplementary Data 2). 57 metabolic enzymes were identified to be required for tumor cell survival in confining capillaries (Fig. 1b, c). 10 candidate genes were selected for further investigation, including kynureninase (KYNU), phenylalanine hydroxylase (PAH), phosphatidylinositol-4-phosphate 3-kinase C2 domain-containing subunit alpha (PIK3C2A), myotubularin-related protein 8 (MTMR8), 6-pyruvoyltetrahydropterin synthase (PTS), diacylglycerol kinase iota (DGKI), L-histidine phosphatase (LHPP), aldehyde dehydrogenase 1 family member B1 (ALDH1B1), retinol dehydrogenase 12 (RDH12) and methylthioribose-1-phosphate isomerase 1 (MRI1) (Supplementary Fig. 1c).

We next evaluated the association of these 10 candidate genes with lung cancer metastasis by using the Gene Expression Omnibus (GEO) dataset (GSE72094), which showed that PTS, DGKI, and ALDH1B1 mRNA levels were strongly correlated with the malignancy and prognosis of lung cancer patients (Supplementary Fig. 1d, e). Furthermore, we defined the relationship between these 3 genes and metastatic recurrence by utilizing the GEO dataset (GSE37745). Among them, ALDH1B1 expression was higher in patients with metastatic recurrence compared to those without metastatic recurrence, while PTS and DGKI showed no differences in their expression between these patients (Fig. 1d and Supplementary Fig. 1f), suggesting that ALDH1B1 may play an important role in tumor metastasis. To investigate whether ALDH1B1 promotes tumor cell survival in confining spaces, we simulated confinement scenario in vitro by adopting a transwell chamber with 8 μm pores (Supplementary Fig. 2a). We depleted ALDH1B1 with short hairpin RNA (shRNA) in A549 cells (Supplementary Fig. 2b), and found that ALDH1B1 depletion increased tumor cell death under confinement but did not significantly affect cell survival in unconfinement conditions (Supplementary Fig. 2c). In addition to ALDH1B1, we identified ALDH9A1 and ALDH6A1 as secondary candidates among the 18 ALDH isoforms in our CRISPR-based screen (Supplementary Fig. 2d). We established A549 cells with ALDH6A1 or ALDH9A1 depletion (Supplementary Fig. 2e, f) and found that ALDH9A1 depletion increased tumor cell death in confining pores (Supplementary Fig. 2g), though its effect was weaker than that of ALDH1B1 ablation. In contrast, ALDH6A1 depletion showed no significant effects on confined tumor cell survival (Supplementary Fig. 2h). Transcriptomic analysis of the GSE288929 dataset further demonstrated that baseline ALDH1B1 expression exceeded ALDH9A1 levels in A549 cells (Supplementary Fig. 2i), potentially explaining the dominant functional role of ALDH1B1 in confining spaces. Collectively, among ALDH isoforms, ALDH1B1 is identified as the primary regulator governing tumor cell survival under spatial confinement.

During metastatic dissemination, tumor cells must navigate two critical physical constraints, including narrow capillary lumens in distant organs and endothelial gaps during intravasation and extravasation[3]. To further confirm that ALDH1B1 contributes to tumor cell survival in the vasculature of distant organ, we implanted luciferase-expressing A549 cells with or without ALDH1B1 into female BALB/c athymic nude mice via tail vein injection. At 0.5 h post tail vein injection, bioluminescence imaging of the mice showed comparable tumor cell signals between the two groups. At 24 h post-injection, the ALDH1B1-depleted group showed accelerated diminishment of bioluminescence signals from tumor cells in lung tissues (Fig. 1e). Moreover, these mice were sacrificed and lung tissues were dissected at 24 h after injection. TUNEL assay on dissected lung tissues showed that ALDH1B1 depletion increased tumor cell death in lung capillaries (Fig. 1f). In addition, we also investigated whether ALDH1B1 regulates tumor cell survival during extravasation through endothelial gaps. Most surviving tumor cells begin extravasation between 24 and 36 h after intravenous injection[20–22]. Since visualizing the survival of cells trapped within endothelial gaps proved challenging, we instead quantified both the cells remaining in capillaries and those that had completed extravasation 48 h after injection, when most cells completed extravasation. Immunofluorescence analysis showed that ALDH1B1 depletion led to a twofold reduction in tumor cells retained within capillaries and a fivefold decrease in extravasated tumor cells (Fig. 1g), suggesting that ALDH1B1 likely also contributes to the survival of tumor cells during extravasation.

Furthermore, we implanted these cells into female BALB/c athymic nude mice via intracardiac injection. Bioluminescence imaging of the mice showed that ALDH1B1 depletion accelerated the diminishment of bioluminescence signals of tumor cells in brain 7 days after inoculation, and dampened brain metastasis of the cells 63 days after inoculation (Fig. 1h and Supplementary Fig. 2j). Hematoxylin eosin (H&E) staining of dissected brain tissues confirmed that ALDH1B1 depletion reduced the number and size of brain metastatic lesions (Fig. 1i). Consequently, the mice bearing with ALDH1B1-depleted

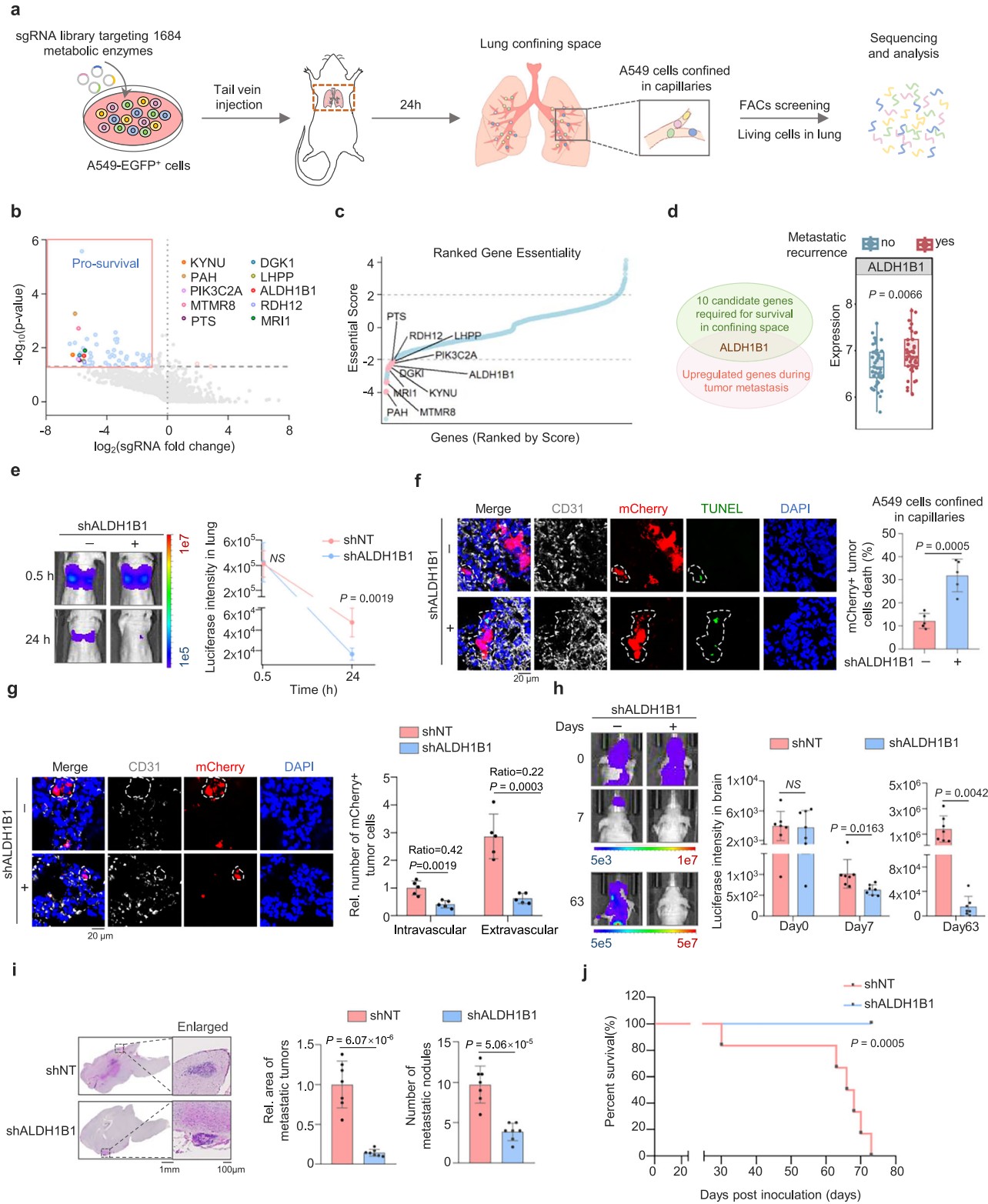

tumors survived longer (Fig. 1j). These results demonstrate that ALDH1B1 is required for tumor metastasis at least partially by enhancing tumor cell survival within capillaries and during extravasation.

## ALDH1B1 sustains tumor cell survival in confining spaces by detoxifying aldehydes

To investigate how ALDH1B1 promotes tumor cell survival in confining spaces, we depleted ALDH1B1 in A549, H1299 cells, or H460 cells and rescued the cells with wild-type ALDH1B1 (WT) or ALDH1B1 enzymatic-

dead (ED) mutant, which indeed had much lower activity than WT (Supplementary Fig. 3a–d). While rescued expression of ALDH1B1 WT almost completely restored the survival and migratory capability of ALDH1B1-depleted cells in confining pores, excluding the off-target possibility of ALDH1B1 shRNA. However, tumor cells rescued with ALDHB1 ED could not recover the survival and the migratory capability of ALDH1B1-depleted cells in confinement (Fig. 2a, b and Supplementary Fig. 3e–h). We also examined the survival of tumor cells in poly-dimethylsiloxane (PDMS) microchannels and found that ALDH1B1

**Fig. 1 | ALDH1B1 is required for tumor cell survival in confining space and distant metastasis. a** Schematic of the in vivo metabolic enzyme screen, using sequencing of viable tumor cells from lung-confining capillaries to identify essential genes. The cell isolation procedure is detailed in Supplementary Fig. 1a. **b, c** Volcano plots show altered genes ($P < 0.05$ and $\log_2$ sgRNA fold change (FC) $\geq 1$ or $\leq -1$) in colored blue (pro-survival) or pink (pro-death). Ten candidates satisfied criteria ($\log_2$ FC $\leq -2$, good sgRNA count $\geq 5$, $P < 0.05$) are labeled (**b**) and highlighted in pink (**c**) ($n = 3$ independent experiments). **d** Venn diagram of ten candidate genes associated with lung cancer metastasis (left). Box plots show ALDH1B1 expression in lung cancer patients with ($n = 49$) and without ($n = 47$) metastatic recurrence (GEO: GSE37745). The center line represents the median, box limits the upper and lower quartiles, and whiskers the minimum and maximum values (right). **e, f** Luciferase/mCherry-A549 cells expressing shNT or shALDH1B1 were tail-vein injected into mice. Representative bioluminescence images from $n = 5$ mice per group (**e**, left) and quantification (**e**, right). Representative images of dead tumor cells within capillaries from $n = 5$ mice per group (white) (**f**, left) and quantified

percentages of dead tumor cells (**f**, right). DAPI, 4′,6-diamidino-2-phenylindole. **g** mCherry-expressing A549 cells expressing shNT or shALDH1B1 were tail-vein injected into mice. Representative images show tumor cells (red) localized outside capillaries from $n = 5$ mice per group (white) (left). The number of intravascular and extravascular tumor cells was quantified (right). **h, i** A549-luciferase cells expressing shNT or shALDH1B1 were intracardially injected into mice. Representative bioluminescence images from $n = 7$ mice per group (**h**, left) and quantification (**h**, right). Representative H&E-stained brain sections from $n = 7$ mice per group (**i**, left) and quantification of tumor areas and numbers (**i**, right). **j** Kaplan–Meier survival analysis of mice intracardially injected with A549 cells expressing shNT or shALDH1B1 ($n = 6$ per group). Data are presented as mean ± SD per mouse (**e–i**). $P$-value for each sgRNA was calculated using two-tailed Student's $t$ test, whereas $P$-value for each gene was calculated using one-tailed Student's $t$ test (negative selection) (**b, c**). $P$-values were calculated using unpaired two-tailed Student's $t$ test (**d–i**) and two-tailed log-rank test (**j**). NS not significant, Rel. relative. Source data are provided as a Source Data file.

depletion increased tumor cell death in these channels (Supplementary Movies 1–4). These results indicate that ALDH1B1 activity was essential for tumor cell survival and subsequent migration in confining spaces.

ALDH1B1 has previously been reported to metabolize aliphatic aldehydes, aromatic aldehydes, and the products of lipid peroxidation, 4-hydroxynonenal and malondialdehyde. Moreover, ALDH1B1 exhibited a higher affinity for short-chain and medium-chain aliphatic aldehydes than other type aldehydes[13]. We thus examined the levels of ALDH1B1 substrate aldehydes, including acetaldehyde, propionaldehyde, hexanal, and nonanal. The levels of these aldehydes were increased in confined cells and were further upregulated in the cells after ALDH1B1 depletion (Fig. 2c and Supplementary Fig. 3i). We further treated tumor cells expressing shNT or shALDH1B1 in confinement with or without aldehyde scavenger Acloproxalap[23]. Acloproxalap almost completely recovered the survival and the migratory capability of ALDH1B1-depleted confined cells (Fig. 2d, e and Supplementary Fig. 3j, k). Of note, wound healing assay showed that ALDH1B1 depletion did not influence the mobility of these cells under no confinement (Supplementary Fig. 4a, b). Similarly, both the proliferation and the survival of tumor cells were not influenced by ALDH1B1 depletion under no confinement (Supplementary Fig. 4c–e). These results demonstrate that ALDH1B1-catalyzed aldehydes detoxification is required for tumor cell survival in confinement.

Ferroptosis is a non-apoptotic cell death mechanism characterized by iron-dependent membrane lipid peroxidation[24,25]. Recently, accumulating evidence indicates that aldehydes can damage ferroptosis-associated proteins and exacerbate intracellular oxidative stress and damage to the cell membrane, accelerating lipid peroxidation and leading to ferroptosis[26,27]. To investigate how ALDH1B1 regulates the survival of confined cells, we thus tested whether ALDH1B1 regulates the ferroptosis. Confinement significantly increased lipid peroxidation in tumor cells, and depletion of ALDH1B1 further exacerbated these effects (Supplementary Fig. 5a, b). Additionally, ALDH1B1 depletion decreased the GSH/GSSH ratio in confined cells, while Acloproxalap greatly abrogated these changes in ALDH1B1-depleted cells (Fig. 2f, g and Supplementary Fig. 5c, d). Ferroptosis is normally inhibited by the continuous activity of coupled enzyme-metabolite systems that prevent the accumulation of membrane lipid peroxides to toxic levels. The enzyme glutathione peroxidase 4 (GPX4) reduces potentially toxic lipid hydroperoxides to less dangerous lipid alcohols. GPX4 appears to be the most critical enzyme for preventing lipid hydroperoxide accumulation in most cells[28–30]. Thus, we overexpressed GPX4 in ALDH1B1-depleted cells (Supplementary Fig. 5e, f). GPX4 overexpression recovered the survival and the migration of ALDH1B1-depleted cells in confining spaces (Fig. 2h, i and Supplementary Fig. 5g, h). 4-Hydroxynonenal (4-HNE) is a well-established marker of aldehyde metabolism and a known substrate of ALDH1B1[13,31],

directly reflecting intracellular aldehyde accumulation and lipid peroxidation[32]. Immunofluorescence analysis of mouse lung tissues harvested 24 h after tail vein injection showed that ALDH1B1 depletion significantly increased 4-HNE levels in tumor cells confined within lung capillaries (Fig. 2j). In addition, given that aldehydes also induce apoptosis[33], we examined whether ALDH1B1 regulates apoptosis under confinement by measuring caspase-3/7 activity. Caspase-3/7 activity was elevated in confined cells, and this effect was further exacerbated by ALDH1B1 depletion (Supplementary Fig. 5i). However, compared to its effect on ferroptosis, ALDH1B1 depletion had a much weaker influence on the apoptosis of confined tumor cells. These results are consistent with previous studies demonstrating that aldehydes induce both apoptosis and ferroptosis[26,33], and that aldehydes derived from ALDH inhibition tend to promote ferroptosis[20,27,34]. Collectively, these results suggest that ALDH1B1 plays a dual protective role in confined tumor cells by suppressing both ferroptosis and apoptosis, with its anti-ferroptotic effect being mechanistically dominant.

Given the necessity of ALDH1B1 activity for the survival of the confined cells, we next evaluated the therapeutic potential of ALDH1B1 inhibitor for lung cancer metastasis. Firstly, we treated A549 or H1299 cells with ALDH1B1 inhibitor, IGUANA-1[35]. Transwell migration assay showed that IGUANA-1 suppressed the confined migration of these cells in a dose-dependent manner without influencing their proliferation (Fig. 2k and Supplementary Fig. 5j–l). Moreover, we implanted luciferase-expressing A549 cells into randomized female BALB/c athymic nude mice via intracardiac injection, followed by the administration of IGUANA-1. Bioluminescence imaging of the mice showed that IGUANA-1 treatment dampened brain metastasis of lung cancer cells (Fig. 2l and Supplementary Fig. 5m). H&E staining of dissected brain tissues confirmed that IGUANA-1 reduced the number and size of brain metastatic lesions (Fig. 2m). Similarly, the survival duration of tumor-bearing mice was extended by IGUANA-1 treatment (Fig. 2n).

## Compressive force induces ALDH1B1 upregulation in confined cells

To investigate how ALDH1B1 is regulated in tumor cells during confined migration, we examined ALDH1B1 expression in confined cells (Fig. 3a). The mRNA and protein levels of ALDH1B1 were increased in confined cells (Fig. 3b, c and Supplementary Fig. 6a, b). Cells are subjected to compressive force in confined environments[36]. Multiple mechanosensitive calcium channels, including Piezo-type mechanosensitive ion channel (PIEZO), Orai calcium-release-activated calcium modulator (ORAI), and transient receptor potential vanilloid (TRPV) members, play important roles in cancer biology, though their activation mechanisms differ significantly[37]. PIEZO1 is a well-characterized mechanosensitive ion channel that directly converts extracellular mechanical forces such as shear stress and membrane stretch into $Ca^{2+}$ signals through plasma membrane deformation[38–40]. ORAI1 mediates

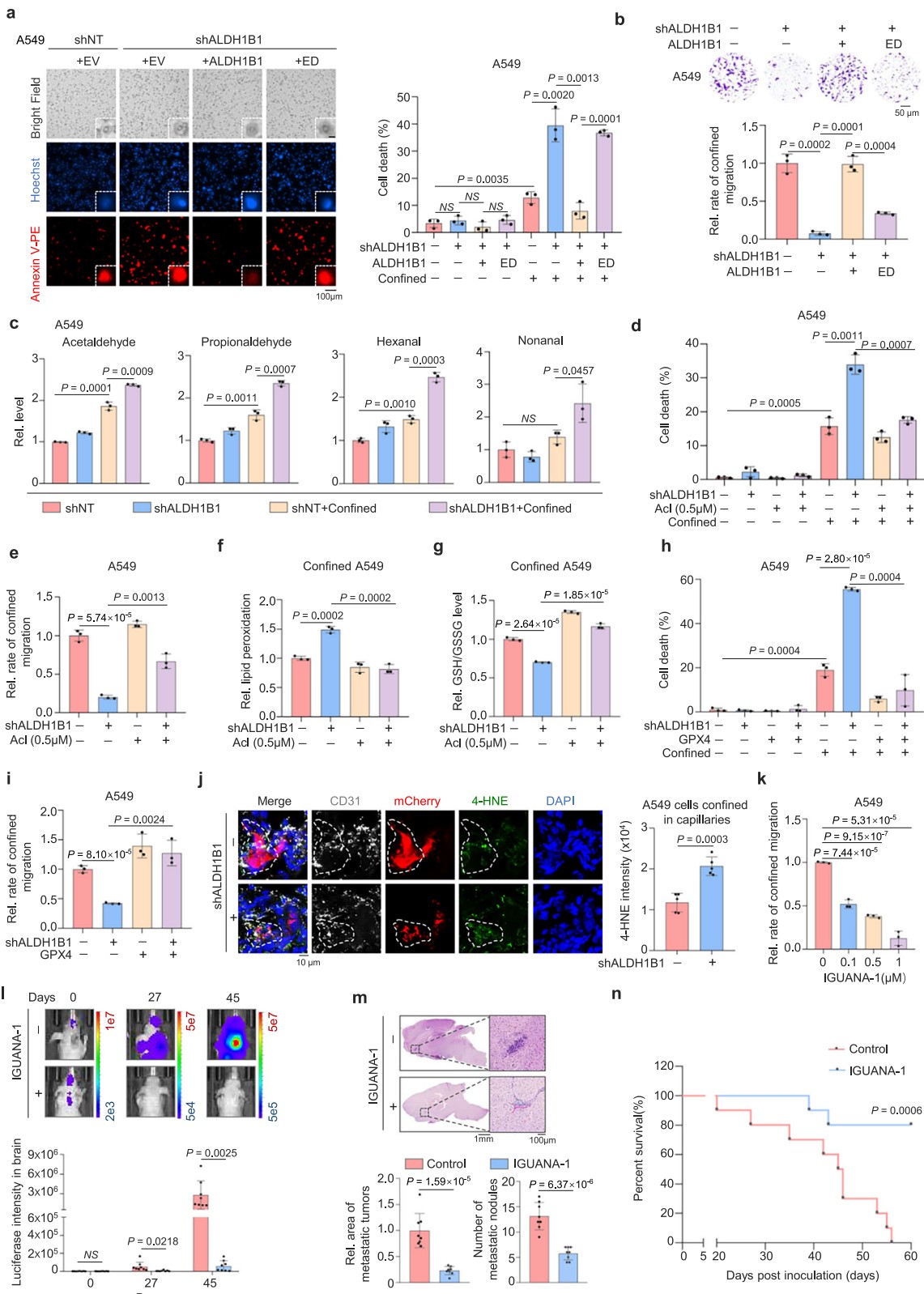

store-operated calcium entry (SOCE) and is activated indirectly upon endoplasmic reticulum (ER) calcium depletion via stromal interaction molecule 1 (STIM1) coupling, rather than through direct mechanical stimulation[41–43]. TRPV4, meanwhile, functions as a polymodal sensor that responds to both mechanical and chemical stimuli[44,45]. Among these, PIEZO1 stands out as the primary channel responsible for rapid, force-dependent calcium signaling in response to physical

perturbation. We thus depleted PIEZO1 in A549 and H1299 cells (Supplementary Fig. 6c, d). PIEZO1 depletion abrogated the upregulation of ALDH1B1 mRNA and protein levels in confined cells (Fig. 3d, e and Supplementary Fig. 6e, f). Meanwhile, we also depleted ORAI1 or TRPV4 in A549 cells (Supplementary Fig. 6g, h). In contrast, ORAI1 depletion had no significant effect on ALDH1B1 expression in confined tumor cells (Supplementary Fig. 6i, j), while TRPV4 depletion partially

**Fig. 2 | ALDH1B1 sustains tumor cell survival in confining spaces by detoxifying aldehydes. a, b** ALDH1B1-depleted A549 cells were reconstituted with ALDH1B1 WT or ED. After transwell migration, cells were stained with Annexin V-PE. Representative images of dead confined cells from $n = 3$ biologically independent experiments (**a**, left) and the quantification of cell death (**a**, right). Scale bar represents 15 μm (zoomed-in images). Transwell migration assays show representative images from $n = 3$ biologically independent experiments (**b**, upper) and quantification of migrated cells (**b**, lower). **c** A549 cells expressing shNT or shALDH1B1 with or without confinement were collected for aldehyde level measurement. **d–g** ALDH1B1-depleted A549 cells were treated with or without Acloproxalap (Acl). Quantification of unconfined and confined cell death (**d**) and of migrated cells (**e**) are shown. Lipid peroxidation in cells was assessed by BODIPY 581/591 C11 staining (**f**). The glutathione (GSH)/oxidized glutathione (GSSG) ratio in cells was quantified (**g**). **h, i** ALDH1B1-depleted A549 cells were overexpressed with empty vector (EV) or GPX4. Quantification of unconfined and confined cell death (**h**) and of migrated cells (**i**) are shown. **j** A549 cells expressing mCherry and shNT or shALDH1B1 were tail-vein injected into mice. Representative images of tumor cells (red) constrained in lung capillaries (white) from $n = 5$ mice per group (left) and scoring of 4-HNE intensity (right) were quantified. **k** A549 cells were treated with IGUANA-1 and subjected to transwell migration assays. Statistical analyses of the migrated cells are presented. **l–n** Luciferase-expressing A549 cells were intracardially injected into mice and treated with IGUANA-1. Representative bioluminescence images from $n = 8$ mice per group (**l**, upper) and quantification (**l**, lower) are shown. Representative images of H&E-stained brain sections from $n = 8$ mice per group (**m**, upper) and metastatic tumor areas and numbers were quantified (**m**, lower). Kaplan–Meier survival analysis of mice intracardially injected with A549 cells treated with or without IGUANA-1 ($n = 10$ per group) (**n**). Data are presented as mean ± SD per mouse (**j, l, m**). Data are presented as mean ± SD ($n = 3$ biologically independent experiments) (**a–i, k**). P-values were calculated using unpaired two-tailed Student's $t$ test (**a–m**) and two-tailed log-rank test (**n**). NS not significant, Rel. relative. Source data are provided as a Source Data file.

abrogated ALDH1B1 upregulation (Supplementary Fig. 6k, l), though unlike PIEZO1 depletion, which completely abrogated ALDH1B1 upregulation in confined tumor cells. These results suggest that under spatial confinement, PIEZO1 may serve as the primary mediator, and implicate that mechanical force drives ALDH1B1 upregulation in confined tumor cells.

To test the hypothesis, we applied direct compressive force to tumor cells and examined ALDH1B1 expression, which showed that the mRNA and protein levels of ALDH1B1 were upregulated in the cells after compression in a dose and time-dependent manner (Fig. 3f–h and Supplementary Fig. 6m–t). We also examined ALDH9A1 expression under compression and revealed modest but statistical upregulation of ALDH9A1 under compression, However, this mechanosensitive upregulation was attenuated compared to ALDH1B1's response (Supplementary Fig. 6u, v). ECM stiffness has been widely utilized as a key biomechanical parameter to mimic tissue-specific mechanical microenvironments in vitro[46,47]. To determine whether matrix stiffness regulates ALDH1B1 expression, we tested distinct ECM stiffness regimes based on reported lung tissue mechanics[48]. We found no stiffness-dependent modulation of ALDH1B1 expression, suggesting that its regulation is independent of matrix stiffness (Supplementary Fig. 6w).

Next, we investigated whether PIEZO1 contributes to compression-induced ALDH1B1 expression. PIEZO1 depletion abrogated the increase in ALDH1B1 mRNA and protein expression in the cells after compression (Fig. 3i, j and Supplementary Fig. 6x, y). Moreover, PIEZO1 depletion increased tumor cell death in confining pores (Fig. 3k). Animal experiments showed that PIEZO1 depletion accelerated the diminishment of bioluminescence signals of tumor cells in the lungs (Fig. 3l). TUNEL assay with these lung tissues showed that PIEZO1 depletion increased tumor cell death in lung capillaries (Fig. 3m). Additionally, PIEZO1 depletion inhibited ALDH1B1 expression in tumor cells confined within lung capillaries (Fig. 3n).

Collectively, these results demonstrate that the expression of ALDH1B1 is increased in tumor cells during confined migration in response to compressive force.

## NF-κB upregulates ALDH1B1 expression upon compression

Given that the mRNA levels of ALDH1B1 is upregulated upon compression, we further explored how the transcription of ALDH1B1 is regulated. We constructed a luciferase reporter system containing ALDH1B1 promoter (ALDH1B1-Luc) and transfected the construct into A549 cells and H1299 cells. As shown in Fig. 4a and Supplementary Fig. 7a, the activity of ALDH1B1 promoter was enhanced in the cells upon compression. To identify the upstream signaling that regulates ALDH1B1 expression, we treated A549 cells with the inhibitors of signal pathways that have been shown to play the crucial role in cancer progression, Ravoxertinib (ERK inhibitor), Mirdametinib (MEK1/2 inhibitor), MK-2206 (AKT inhibitor), Verteporfin (YAP inhibitor), and PDTC (NF-κB inhibitor). Among them, only NF-κB inhibitor abrogated compressive force-induced increase of ALDH1B1 protein levels (Fig. 4b and Supplementary Fig. 7b). Furthermore, the depletion of RelA, the key transcription factor of NF-κB pathway, abrogated compressive force-induced increase in the promoter activity, the mRNA levels and the protein levels of ALDH1B1 (Fig. 4c–f and Supplementary Fig. 7c–f). In addition, the mutation of RelA-binding motif in ALDH1B1 promoter abrogated compression-induced increase in its promoter activity (Fig. 4g and Supplementary Fig. 7g). Chromatin immunoprecipitation (ChIP) assay showed that RelA was recruited to ALDH1B1 promoter upon compression (Fig. 4h and Supplementary Fig. 7h). These results demonstrate that RelA regulates ALDH1B1 promoter activity to increase its expression in tumor cells under compression.

To confirm whether RelA regulates ALDH1B1 expression to detoxify aldehydes in confined cells, we overexpressed ALDH1B1 in RelA-depleted cells (Fig. 4i and Supplementary Fig. 7i). The levels of aldehydes increased in RelA-depleted confined cells, while ALDH1B1 overexpression reduced the levels of the aldehydes in these cells (Fig. 4j). Similarly, lipid peroxidation levels were increased in confined cells after RelA depletion, while ALDH1B1 overexpression almost completely abrogated such increase in RelA-depleted cells (Fig. 4k and Supplementary Fig. 7j). Moreover, the survival and the migratory capability of RelA-depleted cells were also greatly recovered by ALDH1B1 overexpression (Fig. 4l, m and Supplementary Fig. 7k, l), whereas RelA depletion showed no significant impact on lipid peroxidation or survival in unconfined tumor cells (Fig. 4k, l and Supplementary Fig. 7j, k). These results elaborate that NF-κB upregulates ALDH1B1 expression to inhibit ferroptosis in confined cells, thereby promoting confined migration of tumor cells.

## Compressive force activates IKKβ and NF-κB via PIEZO1

In most cell types, NF-κB complexes are retained in the cytoplasm by a family of inhibitory proteins known as inhibitors of NF-κB (IκBs). Activation of NF-κB typically involves the phosphorylation of IκB by the IκB kinase (IKK) complex, which results in IκB degradation. This releases NF-κB and allows it to translocate freely to the nucleus[49]. We next investigated how NF-κB is activated upon compression. Both spatial confinement and mechanical compression greatly increased IKKβ serine (S) 177/181 phosphorylation (pS177/181) that is essential for canonical IKKβ activation as well as the S32/36 phosphorylation and degradation of IκBα (Fig. 5a, b and Supplementary Fig. 8a, b). The levels of nuclear RelA were also increased upon compression (Fig. 5c and Supplementary Fig. 8c). Furthermore, we constructed luciferase reporter system containing the responsive elements of NF-κB and observed that NF-κB was activated after compression (Fig. 5d and Supplementary Fig. 8d). Importantly, we depleted IKKβ in tumor cells (Supplementary Fig. 8e, f) and observed that depletion of IKKβ suppressed the compression-induced phosphorylation and

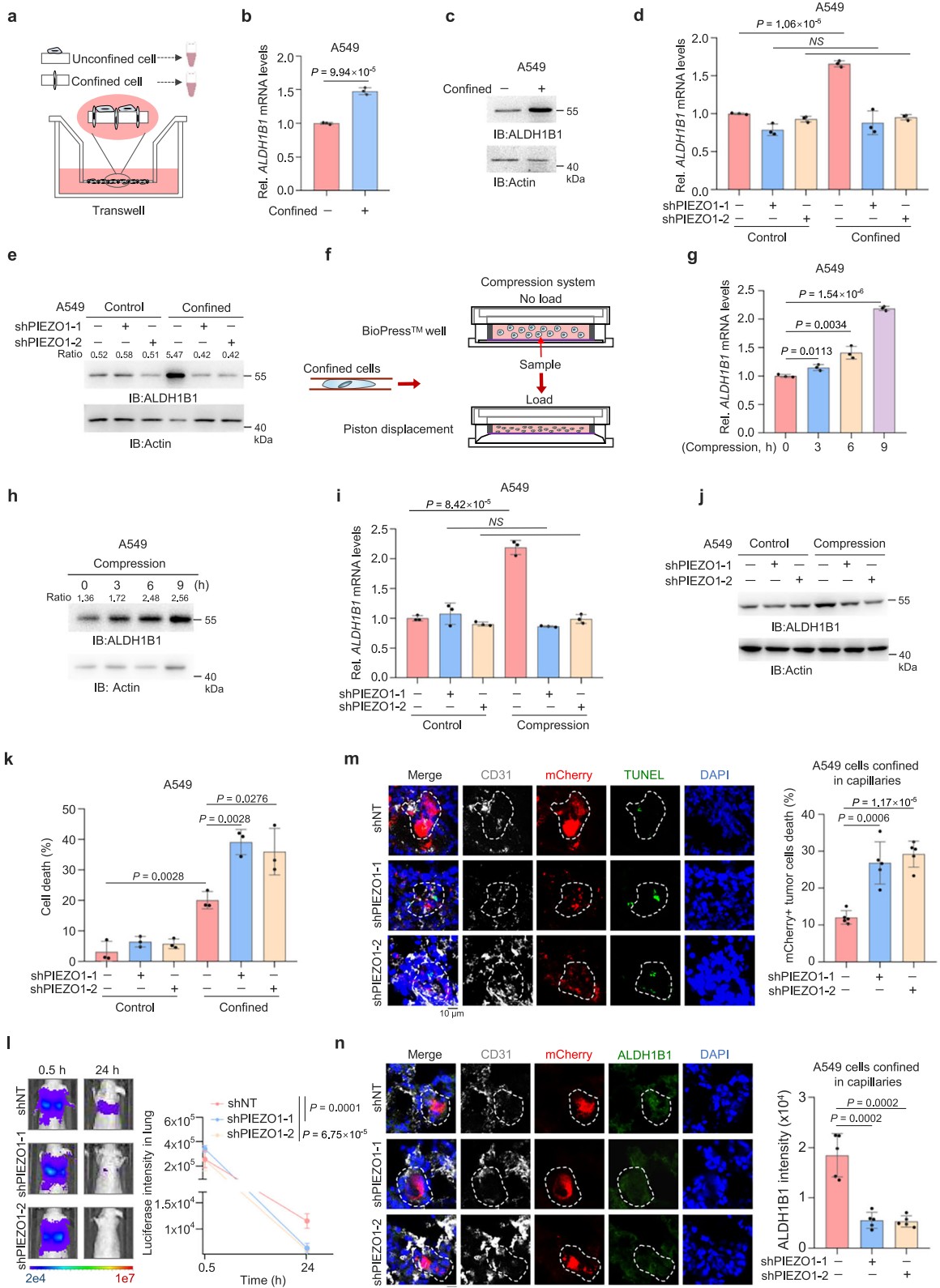

degradation of IκBα as well as ALDH1B1 expression (Fig. 5e–g and Supplementary Fig. 8g–i). In addition, we observed that PIEZO1 depletion greatly inhibited IKKβ pS177/181, IκBα pS32/36, and IκBα degradation in the cells under compression (Fig. 5h and Supplementary Fig. 8j). These results indicate that compressive force activates IKKβ and its downstream NF-κB via PIEZO1, thereby upregulating ALDH1B1 expression.

## CSK23 phosphorylates IKKβ to activate NF-κB

To investigate how compressive force activates IKKβ phosphorylation, we conducted mass spectrometry analyses of IKKβ-associated proteins after compression (Fig. 6a and Supplementary Data 3). Among IKKβ-associated protein kinases, casein kinase II subunit alpha 3 (CSK23) ranked first with a twofold stronger interaction with IKKβ after compression (Fig. 6b). The interaction between CSK23 and IKKβ was

**Fig. 3 | Compressive force induces ALDH1B1 upregulation in confined cells.**
**a** Schematic of the experimental model for collecting confined and unconfined cells from transwell. **b, c** A549 cells were cultured under confinement or unconfinement conditions. *ALDH1B1* mRNA levels were quantified by quantitative PCR (qPCR) (**b**). ALDH1B1 protein levels were detected by immunoblotting (IB) (**c**). **d, e** A549 cells with or without PIEZO1 depletion were cultured under confinement or unconfinement conditions. *ALDH1B1* mRNA levels were analyzed (**d**). ALDH1B1 protein levels were detected, and band intensities were normalized to β-actin (**e**). **f** Schematic of the compression system. **g, h** A549 cells were treated with or without compression (5 kPa) for various time points. *ALDH1B1* mRNA levels were quantified (**g**). ALDH1B1 protein levels were detected, and band intensities were normalized to β-actin (**h**). **i, j** A549 cells with or without PIEZO1 depletion, were treated with or without compression (5 kPa) for 6 h. *ALDH1B1* mRNA levels were detected (**i**). ALDH1B1 protein levels were analyzed (**j**). **k** A549 cells with or without PIEZO1 depletion were stained with Annexin-V-PE after 6 h of transwell migration. The percentages of unconfined and confined cell death were quantified. **l–n** Luciferase and mCherry-expressing A549 cells, with or without PIEZO1 depletion, were injected into mice. Representative bioluminescence images from $n = 5$ mice per group (**l**, left) and quantification (**l**, right) are shown. Representative images of dead tumor cells within lung capillaries (white) from $n = 5$ mice per group (**m**, left) and the quantified percentages of dead tumor cells are shown (**m**, right). Representative images of tumor cells constrained in lung capillaries from $n = 5$ mice per group (**n**, left) and semiquantitative scoring of ALDH1B1 intensity (**n**, right) are shown. Data are presented as mean ± SD per mouse (**l–n**). Data are presented as mean ± SD ($n = 3$ biologically independent experiments) (**b, d, g, i, k**). Immunoblotting experiments were performed with the indicated antibodies. Data are representative of three independent experiments (**c, e, h, j**). *P*-values were calculated using unpaired two-tailed Student's *t* test (**a, d, g, i, k–n**). NS not significant, Rel. relative. Source data are provided as a Source Data file.

validated by co-immunoprecipitation (Co-IP) experiments (Fig. 6c and Supplementary Fig. 9a). Notably, the depletion of CSK23 resulted in the inhibition of IKKβ pS177/181, IκBα pS32/36, IκBα degradation, and ALDH1B1 expression (Fig. 6d–f and Supplementary Fig. 9b–i). To assess whether CSK23 phosphorylates IKKβ, we performed the in vitro kinase assays by mixing bacterial-purified GST-IKKβ and His-CSK23, which showed that CSK23 indeed directly phosphorylated IKKβ S171/181 (Fig. 6g). Additionally, we also mixed SFB-IKKβ and FLAG-CSK23 immunoprecipitated from A549 cells to conduct in vitro kinase assays, confirming that CSK23 phosphorylated IKKβ at S171/181 (Supplementary Fig. 9j).

In addition, we investigated how CSK23 is activated upon PIEZO1 activation by compressive force. Calcium signals are normally transduced by calcium-sensing messengers such as calmodulin (CaM), calpain, and calcineurin to regulate downstream signaling networks[50]. Among these, CaM emerges as a central $Ca^{2+}$ sensor and molecular integrator, dynamically regulating protein-protein interactions and scaffolding macromolecular complex assembly through its target-binding domains[51,52]. Thus, we tested whether CaM is the calcium messenger between PIEZO1 and CSK23. We treated A549 and H1299 cells with CaM-selective inhibitor W7 and observed that W7 treatment attenuated compression-induced interaction between CSK23 and IKKβ (Supplementary Fig. 9k, l). Meanwhile, we also assessed whether calpain or calcineurin also contributes to PIEZO1-mediated CSK23 activation. We treated A549 cells with the calpain inhibitor MDL-28170 or the calcineurin inhibitor FK506. Unlike W7, MDL-28170 treatment had no significant effect on the compression-induced CSK23-IKKβ interaction, whereas FK506 treatment partially abrogated this interaction (Supplementary Fig. 9m, n). These results suggest that CaM is likely the major calcium messenger between PIEZO1 and CSK23.

## CSK23 is required for the survival of confined cells and distant metastasis

To examine whether CSK23 regulates confined cell survival by activating NF-κB, we overexpressed RelA in CSK23-depleted cells. CSK23 depletion increased lipid peroxidation levels in confined cells, while RelA overexpression completely restored lipid peroxidation levels in CSK23-depleted cells (Supplementary Fig. 10a–e). Consequently, CSK23 depletion increased cell death rate and inhibited the migratory capability of tumor cells in confining spaces, while RelA overexpression restored the survival and migration of CSK23-depleted cells in confinement (Supplementary Fig. 10f–k). Similarly, overexpression of ALDH1B1 decreased lipid peroxidation levels and restored the survival and migration of CSK23-depleted cells in confining spaces (Fig. 6h–j and Supplementary Fig. 10l–s), whereas CSK23 depletion showed no significant impact on lipid peroxidation or survival in unconfined tumor cells (Fig. 6h, i and Supplementary Fig. 10d–h and 10o–q). Moreover, we implanted

CSK23-depleted tumor cells with or without ALDH1B1 overexpression into randomized female BALB/c athymic nude mice via intracardiac injection. CSK23 depletion suppressed tumor cell metastasis. H&E staining of dissected brain tissues further confirmed that CSK23 knockdown significantly reduced both the number and size of brain metastatic lesions. Notably, these effects were effectively rescued by ALDH1B1 overexpression (Fig. 6k, l and Supplementary Fig. 10t). These results indicate that CSK23 is required for the survival of confined cells and tumor metastasis by regulating RelA and ALDH1B1.

CX-4945 (Silmitasertib) is the first orally bioavailable inhibitor of casein kinase 2 (CK2) with acceptable pharmacological properties[53]. We next evaluated the therapeutic potential of Silmitasertib for tumor metastasis. We treated A549 or H1299 cells with Silmitasertib. Transwell migration assay showed that Silmitasertib suppressed the confined migration of these cells in a dose-dependent manner without influencing their proliferation (Fig. 6m and Supplementary Fig. 11a–c). Moreover, we implanted luciferase-expressing A549 cells into randomized female BALB/c athymic nude mice via intracardiac injection, followed by the administration of Silmitasertib. Bioluminescence imaging of the mice showed that Silmitasertib treatment dampened brain metastasis of lung cancer cells (Fig. 6n and Supplementary Fig. 11d). H&E staining of dissected brain tissues further confirmed that Silmitasertib reduced both the number and size of brain metastatic lesions (Fig. 6o). Similarly, the survival duration of tumor-bearing mice was extended by Silmitasertib treatment (Fig. 6p).

## Clinical relevance of NF-κB-dependent ALDH1B1 expression in lung cancer patients

To delineate the clinical relevance of our finding, we examined the levels of ALDH1B1 or the levels of nuclear RelA in primary tumor tissues from lung cancer patients. Immunofluorescence analysis indicated that both ALDH1B1 expression levels and nuclear RelA levels were increased in tumor cells constrained in lung capillaries, compared to tumor cells outside capillaries (Fig. 7a–c). Notably, nuclear RelA levels showed a strong positive correlation with ALDH1B1 expression in confined tumor cells (Fig. 7d). Furthermore, higher 4-HNE levels were observed in intravascular tumor cells compared to extravascular areas (Fig. 7e, f). Importantly, ALDH1B1 expression demonstrated a significant negative correlation with both 4-HNE accumulation and cell death in capillary-constrained tumor cells (Fig. 7g–i). Additionally, capillary-confined tumor cells from patients with metastatic recurrence showed elevated ALDH1B1 expression and reduced 4-HNE accumulation compared to the cells from non-recurrent cases (Fig. 7j, k). These results implicate the clinical relevance of NF-κB-regulated ALDH1B1 expression and cell survival in lung cancer patients and suggest the prognostic potential of ALDH1B1 for patients with metastatic lung cancer.

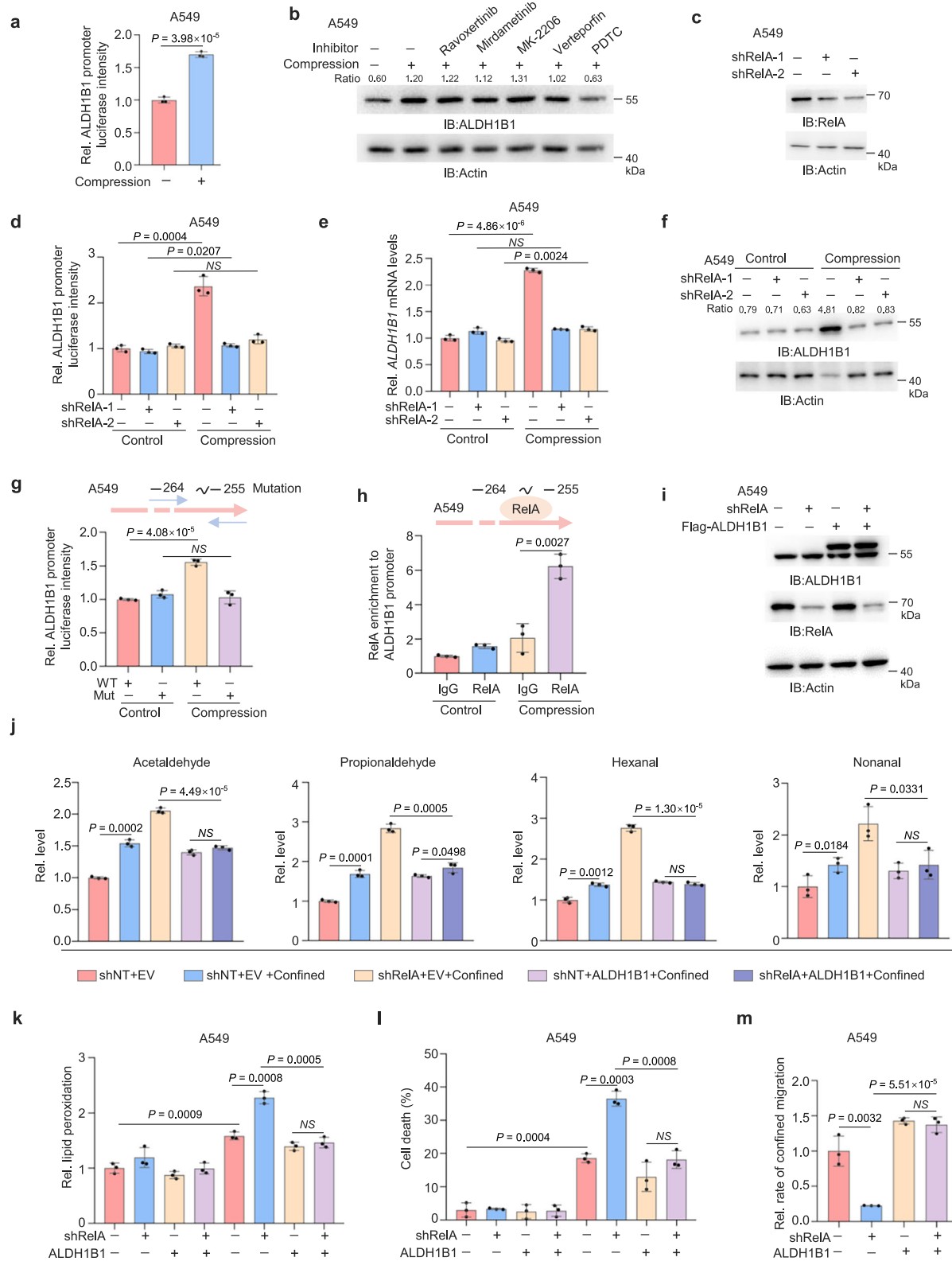

## Discussion

Cancer metastasis is the primary cause of morbidity and mortality in cancer, responsible for approximately 90% of cancer-related deaths[1,2]. Emerging evidence indicates that rewiring metabolism plays a pivotal role in the process of cancer metastasis, including local invasion into the stroma, survival in circulation, and colonization of distant organs[54,55]. The migration of cancer cells is a crucial step that necessitates their navigation through a complex, spatially-confining environment during the cascade of metastasis. However, which metabolic pathway is rewired and how these rewired metabolic pathways facilitate tumor cell migration in spatially-confining environments and subsequent tumor metastasis remain elusive. In this study, we conducted in vivo CRISPR knockout screening in metastasis mouse model of lung cancer by targeting 1685 metabolic enzymes. ALDH1B1

**Fig. 4 | NF-κB upregulates ALDH1B1 expression upon compression. a** A549 cells transfected with ALDH1B1 promoter-luciferase reporter were treated with or without compression (5 kPa) for 6 h. Relative luciferase activity was normalized to controls. **b** A549 cells pretreated with pathway inhibitors: ERK (Ravoxertinib, 10 µM), MEK1/2 (Mirdametinib, 10 µM), AKT1/2/3 (MK-2206, 5 µM), YAP (Verteporfin, 5 µM), NFκB (PDTC, 5 µM) and subjected to compression for 6 h. ALDH1B1 protein levels were assessed and normalized to β-actin. **c** RelA protein levels in A549 cells with RelA depletion were detected. **d** A549 cells with or without RelA depletion, transfected with the ALDH1B1 promoter-reporter, were treated with or without compression for 6 h. Relative luciferase activity was normalized to controls. **e, f** A549 cells with or without RelA depletion treated with or without compression for 6 h. *ALDH1B1* mRNA levels were analyzed (**e**). ALDH1B1 protein levels were detected and normalized to β-actin (**f**). **g** A549 cells transfected with ALDH1B1 promoter-reporter (WT or mutant) were treated with or without compression for 6 h. Relative luciferase activity was normalized to controls. **h** A549 cells treated with or without compression for 6 h were subjected to ChIP assays using indicated antibodies and primers targeting binding sites. Relative DNA levels were normalized to input and immunoglobulin G (IgG) controls. **i** RelA-depleted A549 cells overexpressing EV or ALDH1B1 were analyzed. The samples derive from the same experiment, but different gels for ALDH1B1, Actin, and another for RelA were processed in parallel. **j** RelA-depleted A549 cells overexpressing EV or ALDH1B1 were subjected to transwell migration. Confined and unconfined cells were collected for aldehyde assays. **k–m** RelA-depleted A549 cells overexpressing EV or ALDH1B1 were analyzed after transwell migration. Lipid peroxidation was assessed by BODIPY 581/591 C11 staining (**k**). Cell death percentages were quantified by Annexin V-PE staining (**l**). Migrated cells were quantified (**m**). Data are presented as mean ± SD (*n* = 3 biologically independent experiments) (**a, d, e, g, h, j–m**). Immunoblotting experiments were performed with the indicated antibodies. Data are representative of three independent experiments (**b, c, f, i**). *P*-values were calculated using unpaired two-tailed Student's *t* test (**a, d, e, g, h, j–m**). NS not significant. Rel. relative. Source data are provided as a Source Data file.

was identified to be required for tumor cell survival in confining capillaries. Mechanistically, in response to compressive force, CSK23 interacts with and phosphorylates IKKβ at S177/181 in confined tumor cells, thereby activating NF-κB pathway to upregulate ALDH1B1 expression. Elevated ALDH1B1 expression suppresses ferroptosis by enhancing aldehydes detoxification in confined cells, which supports tumor cell survival during their migration in confining capillaries, thereby promoting lung cancer metastasis (Fig. 8). Notably, ALDH1B1 inhibitor IGUANA-1 dampens brain metastasis of lung cancer and markedly extends the lifespan of metastatic tumors-bearing mice. Our finding reveals the critical role of ALDH1B1 in cancer metastasis and demonstrates the regulatory mechanism of ALDH1B1 expression upon mechanical compression and implicates the therapeutic potential of ALDH1B1 inhibitor to treat metastatic lung cancer.

ALDH1B1, a member of the aldehyde dehydrogenase family, has emerged as a pivotal regulator in tumor biology. It plays well-documented roles in promoting cancer stemness, tumor progression, and therapy resistance across multiple malignancies[56], including colorectal cancer[57–60], pancreatic cancer[61,62], and hepatocellular carcinoma[63], where it drives aggressive phenotypes and correlates with poor clinical outcomes. However, the functional significance of ALDH1B1 in confined migration-a critical mode of in vivo tumor cell locomotion-and its mechanistic role in metastasis remain unclear. In this study, we demonstrate that ALDH1B1 promotes lung cancer metastasis by sustaining tumor cell survival during confined migration. Mechanistically, ALDH1B1 prevents confinement-induced aldehyde accumulation, thereby enhancing the survival of mechanically constrained cells. Notably, ALDH1B1 inhibition effectively suppresses confined migration. Furthermore, immunofluorescence analysis of tumor tissues from lung cancer patients revealed that ALDH1B1 expression was elevated in tumor cells confined within capillaries compared to those outside capillaries, and correlated with metastatic recurrence. Collectively, our study uncovers a unique mechanism of aldehyde metabolism in cancer cells, highlighting its critical role in tumor metastasis.

Current broad-spectrum ALDH inhibitors, such as disulfiram (DSF) and diethylaminobenzaldehyde (DEAB), can overcome functional compensation among isoforms, their clinical translation has been limited by on-target toxicities-including dose-limiting hepatotoxicity, cardiotoxicity, and off-tissue damage to vulnerable cell populations-due to indiscriminate inhibition of multiple ALDH family members[64]. Our multimodal validation establishes ALDH1B1 as a therapeutically viable target for metastatic intervention. Both genetic depletion and pharmacological inhibition of ALDH1B1 significantly reduced metastatic burden and extended median survival in nude mouse models, highlighting its potential to suppress metastatic recurrence in post-operative cancer patients. However, clinical translation requires careful mitigation of on-target toxicities, primarily due to ALDH1B1's physiological roles in intestinal stem cell maintenance and pancreatic regeneration[61,65]. We propose targeting circulating tumor cells during early dissemination-a phase characterized by limited population size and unique metabolic dependencies, which may enable effective metastasis blockade via transient, submaximal ALDH1B1 inhibition. Additionally, ALDH1B1 inhibitors must address the challenge of compensatory activation by other ALDH isoforms, a critical drawback of isoform-selective agents, to enhance therapeutic efficacy and minimize potential resistance mechanisms. To address this, we propose a dual-inhibition strategy targeting both ALDH1B1 and ALDH9A1 may enhance therapeutic efficacy while demonstrating a more favorable toxicity profile compared to pan-ALDH inhibitors.

Although CSK23, a member of the CK2 family, has been associated with lung cancer susceptibility and exhibits debated tumor-suppressive activity in hepatocellular carcinoma[66,67], its broader role in cancer-particularly in metastasis-remains poorly characterized. In our study, we demonstrate the critical role of CSK23 in brain metastasis of lung cancer cells. Pharmacological inhibition by Silmitasertib (CX-4945), a clinically available inhibitor targeting CSK23, significantly reduced metastatic burden and improved survival, suggesting that targeting CSK23 represents a promising strategy to prevent post-surgical metastatic recurrence in lung cancer.

Tumor metabolism is dictated by a variety of intrinsic factors, such as genetic alterations, cell lineage, and histological subtype, and extrinsic factors, such as the access to nutrients and oxygen, the interaction with stromal cells and the exposure to radiation and chemotherapy[68]. Interestingly, emerging studies indicate that tumor cells experience various mechanical stresses from the microenvironment, such as the tension from cell-extracellular matrix interactions, the compressive stress from tumor interior or narrow capillaries and the shear stress within the circulatory system[69]. However, little is known about how metabolic pathways in tumor cells are rewired by mechanical stresses from the microenvironment, especially during metastasis. Here, we demonstrate that in response to compressive force, NF-κB pathway is activated to upregulate ALDH1B1 expression, which suppresses tumor cell ferroptosis during migration in confining capillaries by detoxifying aldehydes, thereby promoting their distant metastasis. Our study reveals the mechanism of compressive force-induced metabolic reprogramming during metastasis.

During metastatic dissemination, tumor cells encounter diverse mechanical stresses in distinct vascular environments. For example, anoikis primarily occurs in large vessels; pulsatile hydrodynamic pressure dominates in arteries; in narrow capillaries, tumor cells mainly experience both fluid shear stress and mechanical compression[70–72]. Our study mainly focuses on mechanical compression, demonstrating that the CSK23-IKKβ-ALDH1B1 pathway is critical for tumor cell adaptation to compressive forces, enabling their survival in constricted capillaries. Targeting this pathway impairs mechanical

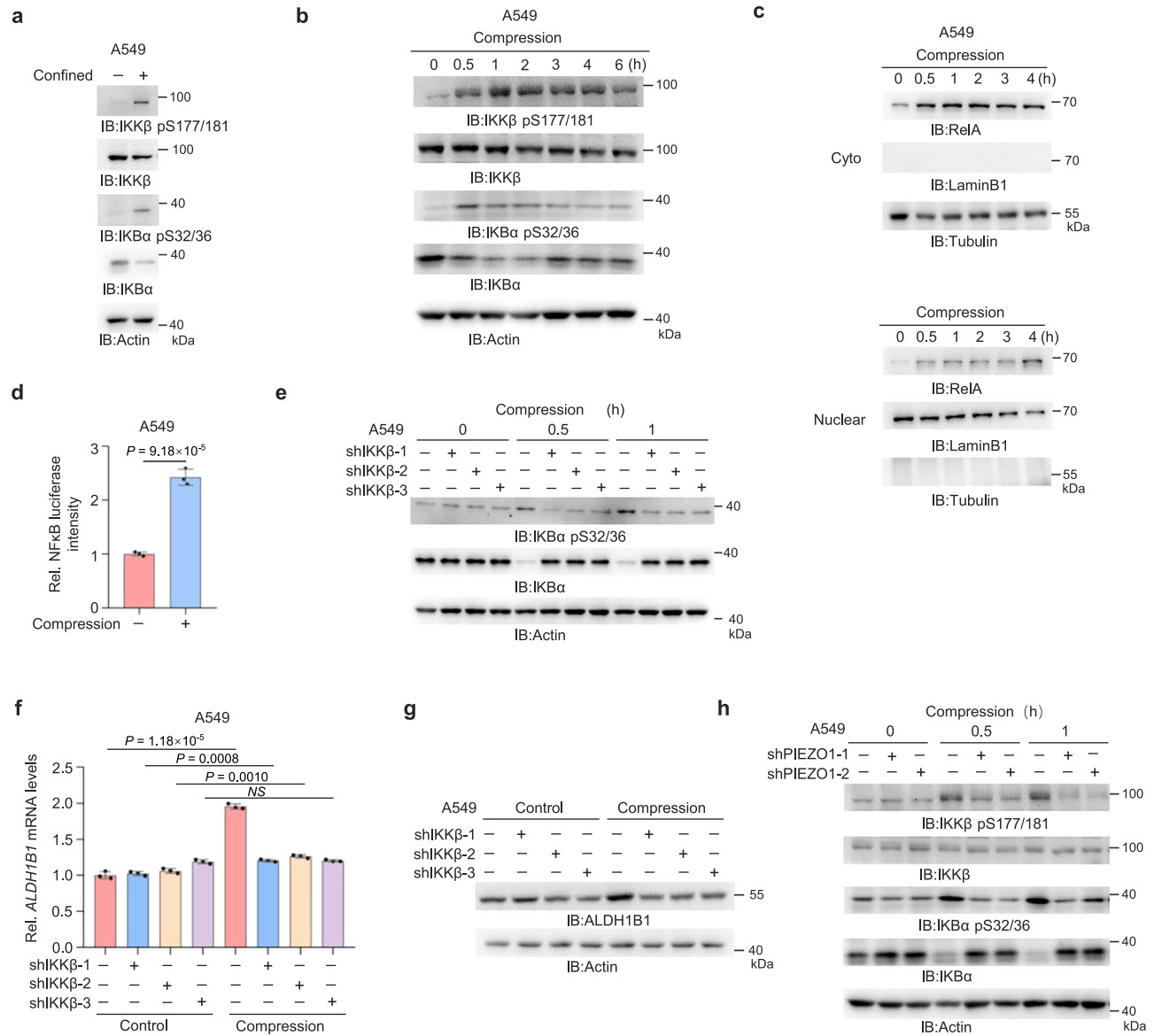

**Fig. 5 | Compressive force activates IKKβ and NF-κB via PIEZO1. a** A549 cells under confinement or unconfinement were analyzed by immunoblotting. The samples derive from the same experiment but different gels for IKKβ, IKBα, and another for IKKβ pS177/181, IKBα pS32/36, Actin were processed in parallel. **b** A549 cells were treated with or without compression at different time and analyzed by immunoblotting. The samples derive from the same experiment, but different gels for IKBα, another for IKKβ pS177/181, IKBα pS32/36, and another for Actin were processed in parallel. **c** A549 cells were treated with or without compression at different time. The cytosolic and nuclear fractions were prepared and analyzed by immunoblotting. The samples derive from the same experiment but different gels for RelA and another for LaminB1, Tubulin were processed in parallel. **d** A549 cells transfected with NF-κB-Luc reporter were treated with or without compression for 2 h. Relative luciferase activity was normalized to controls. **e** A549 cells with or without IKKβ depletion were treated with or without compression.

Immunoblotting analyses were performed. The samples derive from the same experiment, but different gels for IKBα, another for IKBα pS32/36 and another for Actin were processed in parallel. **f, g** A549 cells with or without IKKβ depletion were treated with or without compression for 6 h. *ALDH1B1* mRNA levels were quantified (**f**). ALDH1B1 protein levels were detected (**g**). **h** A549 cells with or without PIEZO1 depletion were treated with or without compression. Immunoblotting analyses were performed. The samples derive from the same experiment, but different gels for IKKβ, IKBα, another for IKKβ pS177/181, IKBα pS32/36, and another for Actin were processed in parallel. Data are presented as mean ± SD (*n* = 3 biologically independent experiments) (**d, f**). Immunoblotting experiments were performed with the indicated antibodies. Data are representative of three independent experiments (**a–c, e, g, h**). *P*-values were calculated using unpaired two-tailed Student's *t* test (**d, f**). NS not significant, Rel. relative. Source data are provided as a Source Data file.

stress adaptation and suppresses distant metastasis. In contrast, fluid shear stress has been extensively studied for its role in tumor metastasis, as it not only disrupts CTCs from settling but may also induce cell-cycle arrest or even cell death. To overcome this mechanical barrier, CTCs enhance their phenotypic plasticity (e.g., through EMT or cytoskeletal remodeling) to improve survival[69,73,74]. Additionally, they form aggregates with platelets, neutrophils, and other cells via adhesive interactions, which protect their surface from shear forces and NK-cell-mediated lysis[75]. Meanwhile, CTCs upregulate bulky glycoproteins

on their surface to increase physical adhesion to endothelial walls, thereby resisting shear forces. These adaptive mechanisms enable tumor cells to overcome shear stress and successfully establish distant metastases[76]. However, whether the CSK23-IKKβ-ALDH1B1 pathway is also critical for tumor cell adaptation to fluid shear stress within capillaries requires further investigation.

The NF-κB transcription factor family is a pleiotropic regulator with indispensable roles not only in inflammatory and immune responses but also in various stages of tumorigenesis and metastatic

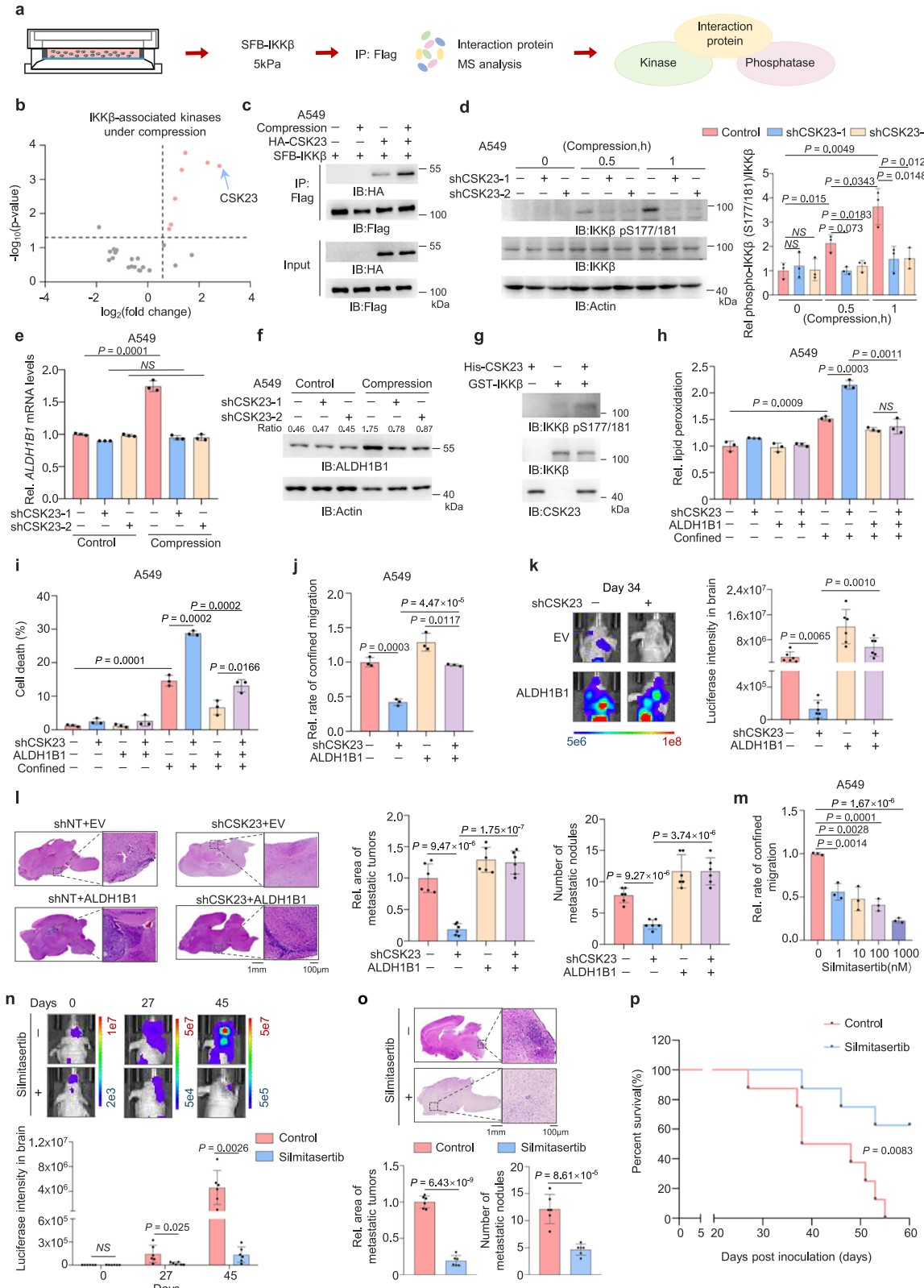

dissemination[77,78]. This factor can be activated through multiple mechanisms in response to diverse stimuli. Cytokines, such as interleukin-1 beta (IL-1β), tumor necrosis factor-alpha (TNF-α), and Interleukin-6 (IL-6), can activate the IκB kinase (IKK) complex by binding to cell surface receptors. This leads to the phosphorylation and degradation of IκB proteins, enabling the activation and translocation of NF-κB into the cell nucleus[79–81]. Metabolic stress also activates

NF-κB through multiple pathways, high glucose activates NF-κB through PKC or reactive oxygen species (ROS) in mesangial cells[82,83]. We previously showed that NF-κB is activated by α-KG in an IKKβ pS177/181-independent manner under low glucose[84]. However, it is mostly unknown whether and how NF-κB is activated by mechanical force. Here, we demonstrate that in response to compressive force, CSK23 interacts with and phosphorylates IKKβ at S177/181 in confined

**Fig. 6 | CSK23 promotes survival of confined cells and distant metastasis by phosphorylating IKKβ. a** Schematic of mass spectrometry analysis for IKKβ-associated kinases or phosphatases under compression. **b** Volcano plot of IKKβ-associated kinases in A549 cells with or without compression for 1 h ($n = 2$ independent experiments). Candidate interactors defined as $P < 0.05$ and FC > 1.5. **c** A549 cells co-expressing HA-CSK23 and SFB-IKKβ under compression were immunoprecipitated. **d** CSK23-depleted A549 cells under compression analyzed by immunoblotting (left) with semi-quantification (right). The samples derive from the same experiment but different gels for IKKβ and another for IKKβ pS177/181, Actin were processed in parallel. **e, f** CSK23-depleted A549 cells under compression, *ALDH1B1* mRNA levels were quantified (**e**). ALDH1B1 protein levels were detected and normalized to β-actin (**f**). **g** In vitro kinase assay using purified GST-IKKβ and HIS-CSK23. The samples derive from the same experiment, but different gels for IKKβ pS177/181 and another for IKKβ, CSK23 were processed in parallel. **h–j** CSK23-depleted A549 cells were overexpressed with EV/ALDH1B1. Lipid peroxidation was assessed (**h**). Cell death were quantified (**i**). Migrated cells were quantified (**j**). **k, l** CSK23-depleted A549-luciferase cells overexpressing EV/ALDH1B1 were intracardially injected into mice. Representative bioluminescence images from $n = 6$

mice per group (**k**, left) and quantification (**k**, right). Representative images of H&E-stained brain sections from $n = 6$ mice per group (**l**, left) and metastatic tumor areas and numbers were quantified (**l**, right). **m** A549 cells treated with Silmitasertib were subjected to transwell migration assays. **n, o** Luciferase-expressing A549 cells were intracardially injected into mice treated with Silmitasertib. Representative bioluminescence images from $n = 6$ mice per group (**n**, upper) and quantification (**n**, lower). Representative images of H&E-stained brain sections from $n = 6$ mice per group (**o**, upper) and metastatic tumor areas and numbers were quantified (**o**, lower). **p** Kaplan–Meier survival analysis of mice intracardially injected with A549 cells treated with or without Silmitasertib ($n = 8$ per group). Data are presented as mean ± SD per mouse (**k, l, n, o**). Data are presented as mean ± SD ($n = 3$ biologically independent experiments) (**d, e, h, i, j, m**). Immunoblotting experiments were performed with the indicated antibodies. Data are representative of three independent experiments (**c, d, f, g**). *P*-values were calculated using two-tailed Student's *t* test (**b**), unpaired two-tailed Student's *t* test (**d, e, h–o**), and two-tailed log-rank test (**p**). NS not significant. Rel. relative. Source data are provided as a Source Data file.

tumor cells, thereby activating NF-κB pathway to promote tumor metastasis. Notably, the CSK23 inhibitor Silmitasertib dampens brain metastasis of lung cancer and markedly extends the lifespan of metastatic tumors-bearing mice, implicating the therapeutic potential of CSK23 inhibitors to inhibit the metastatic recurrence of lung cancer patients after radical surgeries.

## Methods

### Human specimens
The use of human lung tumor specimens and associated data was approved by the Institutional Review Boards of Shanghai Chest Hospital (KS(Y)23082) and Shanghai XinHua Hospital (XHEC-C-2025-155-1), and informed consent was obtained from all patients. No sex and gender-based analysis was performed. The study focused on characterizing the pre-existing pathological phenomenon of tumor cells within confining vasculature, an objective cellular event that is not contingent on patient sex or gender.

### Mice
Six-week-old female BALB/c athymic nude mice were purchased from Lingchang Biotech (Shanghai, China). Littermates were randomly assigned to experimental groups. Sex was not considered as a biological variable in the study design. This is because the primary objective was to investigate the intrinsic behavior of A549 tumor cells under confinement in vivo. The use of a single-sex cohort was chosen to minimize experimental variability and establish a clear baseline for assessing the effects on tumor cell survival and metastasis in this specific context. Mice were housed under specific pathogen-free conditions with a standard chow diet and maintained at a room temperature of 20–26 °C with 40–70% humidity on a 12-h light/12-h dark cycle. All experimental protocols were approved by the IACUC of the Shanghai Institute of Biochemistry and Cell Biology (approval number: SIBCB-S355-2306-18) and complied with all relevant ethical regulations. Throughout the study, no tumor exceeded the IACUC-permitted maximum size of 1.5 cm in diameter.

### Antibodies
Rabbit polyclonal anti-ALDH1B1 (15560-1-AP, Proteintech Group); Rabbit polyclonal anti-CSK23 (PA5-98299, Thermo Fisher Scientific); Rabbit monoclonal anti-Flag (20543-1-AP, Proteintech Group); Rabbit monoclonal anti-HA (3724, Cell Signaling Technology); Mouse monoclonal anti-Actin (60008-1-lg, Proteintech Group); Mouse monoclonal anti-Tubulin (T9026, Sigma); Mouse IgG antibody (sc-2025, Santa Cruz Biotechnology); Rabbit monoclonal anti-RelA (8242, Cell Signaling Technology); Mouse monoclonal anti-RelA (66535-1-Ig, Proteintech Group); Rabbit monoclonal anti-IKKβ (8943, Cell Signaling

Technology); Mouse monoclonal anti-IκBα S32/36 phosphorylation (9246, Cell Signaling Technology); Rabbit monoclonal anti-IKKβ S177/181 phosphorylation (2697, Cell Signaling Technology); Rabbit monoclonal anti-IκBα (10268-1-AP, Proteintech Group); Rabbit monoclonal anti-Lamin B1 (12586 s, Cell Signaling Technology); Goat polyclonal anti-CD31(AF3628, Novus Biologicals); Mouse monoclonal anti-4-HNE (MAB3249, R&D Systems); Goat-anti-mouse IgG second antibody (31160, Thermo Fisher Scientific); Goat-anti-rabbit IgG second antibody (31210, Thermo Fisher Scientific); Donkey-anti-mouse Alexa Fluor 488 (A21202, Thermo); Donkey-anti-rabbit Alexa Fluor cy3 (711-165-152, Jackson ImmunoResearch); Donkey-anti-goat Alexa Fluor 647 (A21447, Thermo).

### Regents
The Annexin-V-PE /7-AAD apoptosis detection kit was purchased from BD Biosciences. The DeadEnd Fluorometric TUNEL System was purchased from Promega. The fluorescent lipid peroxidation sensor BODIPY 581/591 C11 was purchased from Invitrogen. The GSH/GSSG assay kit was purchased from Beyotime. The Caspase-3/7 activity assay kit was purchased from Dalian Meilun Biotechnology. The Flag peptide was purchased from Abclonal. All antibiotics, including ampicillin, kanamycin, puromycin, and hygromycin, were purchased from EMD Biosciences. Trizol was purchased from Life Technologies. The In Vitro DNA Transfection Reagent was purchased from Signagen Laboratories. The MEK1/2 inhibitor (Mirdametinib), AKT1/2/3 inhibitor (MK-2206), and NFκB inhibitor (PDTC) were purchased from Selleckchem. The YAP inhibitor (Verteporfin), ERK inhibitor (Ravoxertinib), ALDH1B1 inhibitor (IGUANA-1), CK2α inhibitor (Silmitasertib), CaM inhibitor (W7), Calpain inhibitor (MDL-28170), Calcineurin inhibitor (FK506), and Acloproxalap were purchased from MCE. D-luciferin was purchased from Xenogen. Acetaldehyde-DNPH, Propionaldehyde-DNPH, Hexanal-DNPH, and Nonanal-DNPH reference standards were purchased from Shanghai Titan.

### DNA constructs and mutagenesis
Human ALDH1B1 and GPX4 were PCR-amplified and cloned into pCDH-Flag. Human RelA and CSK23 were PCR-amplified and cloned into pCDH-Flag and pcDNA3.0-HA. Human ALDH1B1 Promoter was PCR-amplified and cloned into pGL3. The mutations of ALDH1B1 E285A and ALDH1B1 promoter were made using the QuikChange II Site-Directed Mutagenesis Kit. The pGIPZ non-targeting shRNA was generated with the control oligonucleotide 5′-CTCGCTTGGGCGAGAGTAA-3′. The pGIPZ human ALDH1B1 shRNA was generated with 5′-CAAAT-TAACTCTTAGAAGA-3′ oligonucleotide targeting the noncoding region of the ALDH1B1 transcript. The pGIPZ human RelA shRNA-1 was generated with 5′-AGGCGAGAGGAGCACAGAT-3′ oligonucleotide and

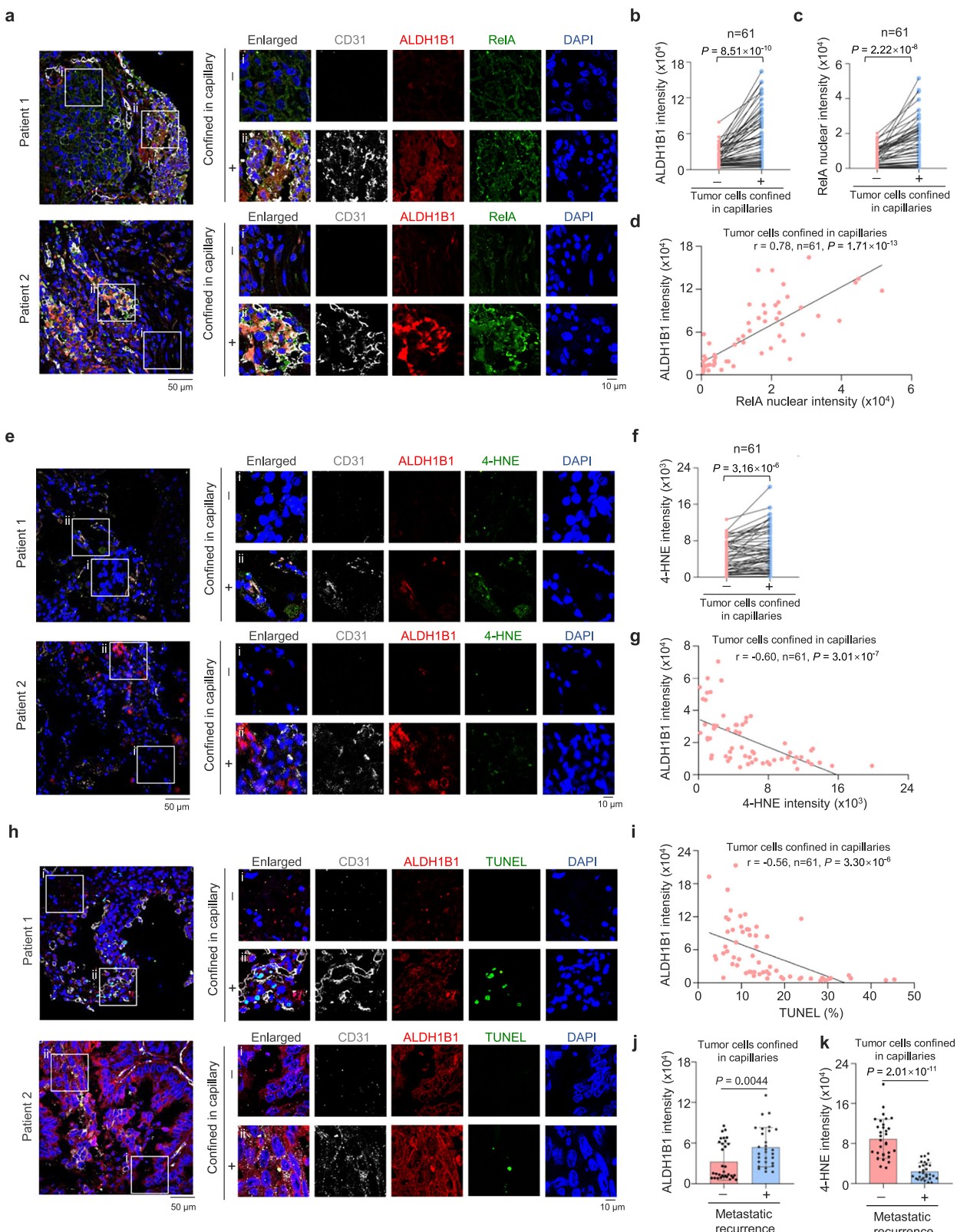

shRNA-2 was generated with 5'-CTCAGTGAGCCCATGGAAT-3' oligo-nucleotide. The pGIPZ human IKKβ shRNA-1 was generated with 5'-GCTTAGAT ACCTTCATGAA-3' oligonucleotide, shRNA-2 was gener-ated with 5'-TGGACATTGTT GTTAGCGA-3' oligonucleotide and shRNA-3 was generated with 5'-AGACCGACATT GTGGACTT-3' oligo-nucleotide. The pGIPZ human CSK23 shRNA-1 was generated with 5'-CAATTGTACCAGACGTTAA-3' oligonucleotide and shRNA-2 was

generated with 5'-CACAGAAAGCTACGACTAA-3' oligonucleotide. The pGIPZ human ALDH9A1 shRNA-1 was generated with 5'-AGGTG-GATCGTTTGGTTAT-3' oligonucleotide and shRNA-2 was generated with 5'-AGGAAAAGAGACTAGGATA-3' oligonucleotide. The pGIPZ human ALDH6A1 shRNA-1 was generated with 5'-AGGTCTT GCTCCGCTATC A-3' oligonucleotide and shRNA-2 was generated with 5'-GCGATCATCCGGACATCAA -3' oligonucleotide. The pGIPZ human

**Fig. 7 | Clinical relevance of NF-κB-dependent ALDH1B1 expression in lung cancer patients. a–d** IF analysis with anti-ALDH1B1, anti-RelA, and anti-CD31 antibodies was performed on lung cancer patients. Images from two representative patients showing tumor cells constrained or unconstrained in lung capillaries from $n = 61$ patients are shown (**a**). Semiquantitative scoring was performed (**b–d**) (paired *t*-test, two-tailed (**b**, **c**); (Pearson product moment correlation test, two-tailed (**d**); $r = 0.78$, $P < 0.0001$). **e–g** IF analysis with anti-ALDH1B1, anti-4-HNE, and anti-CD31 antibodies was performed on the same patients. Images from two representative patients showing tumor cells constrained or unconstrained in lung capillaries from $n = 61$ patients are shown (**e**). Semiquantitative scoring was performed (**f** and **g**) (paired *t*-test, two-tailed (**f**); (Pearson product moment correlation test, two-tailed (**g**); $r = −0.60$, $P < 0.0001$). **h**, **i** IF analysis with anti-ALDH1B1, anti-

CD31 antibodies and TUNEL staining on the same patients. Images from two representative patients showing tumor cells constrained or unconstrained in lung capillaries from $n = 61$ patients are shown (**h**). Semiquantitative scoring was carried out (**i**) (Pearson product moment correlation test, two-tailed; $r = −0.56$, $P < 0.0001$). **j, k** Semiquantitative scoring of ALDH1B1 and 4-HNE intensity in tumor cells confined within capillaries of same patients was calculated. Patients from the cohort ($n = 61$) were categorized based on metastatic recurrence status: with metastatic recurrence ($n = 29$) and without metastatic recurrence ($n = 31$). ALDH1B1 intensity was determined by averaging values from (**d**, **g**, and **i**), while 4-HNE intensity was derived from (**g**). Data represent the mean ± SD for the two groups (two-tailed Student's *t* test). Source data are provided as a Source Data file.

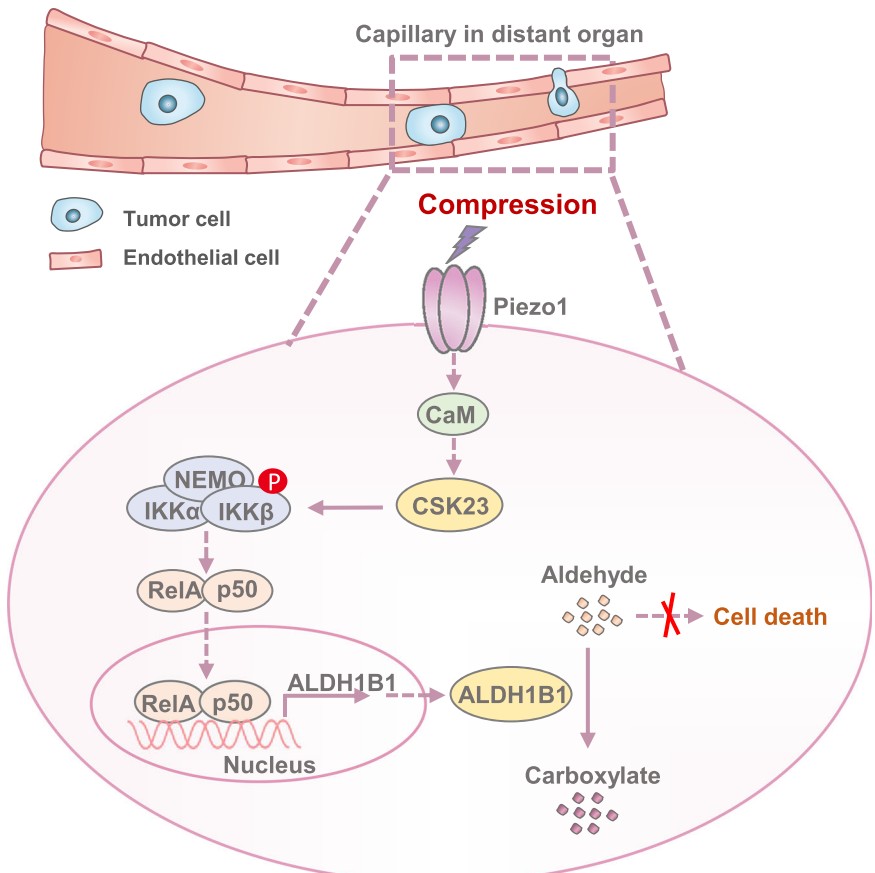

**Fig. 8 | Schematic model for ALDH1B1-promoted tumor metastasis by sustaining cell survival during confined migration.** ALDH1B1 promotes tumor metastasis by sustaining cell survival during confined migration. Mechanistically, in response to compressive force, CSK23 interacts with and IKKβ at S177/181 in confined tumor cells, thereby activating NF-κB pathway to upregulate ALDH1B1

expression. Elevated ALDH1B1 expression protect confined cells from death by enhancing aldehyde detoxification. This mechanism supports tumor cell survival during migration through narrow capillaries, thereby promoting lung cancer metastasis.

TRPV4 shRNA-1 was generated with 5′-ACCAAGTTTGTTACCAAGA-3′ oligonucleotide and shRNA-2 was generated with 5′-TGCTCCTATG-GAGTCACATAA-3′ oligonucleotide. The pGIPZ human ORAI1 shRNA-1 was generated with 5′- ACGCTGCTCTTCCTAGCTG-3′ oligonucleotide and shRNA-2 was generated with 5′-CATGTGTGTGTGACACATA-3′ oligonucleotide. The pLKO.1 human PIEZO1 shRNA-1 was generated with 5′-CTCACCAAGAAGTACAATC AT-3′ oligonucleotide and shRNA-2 was generated with 5′-GCTGCTCTGCTACTTCATC AT-3′ oligonucleotide.

## Cell culture
The human lung cancer cell lines H1299, A549, H460, and the HEK293T cell line were obtained from the Type Culture Collection of the Chinese Academy of Sciences and authenticated by short tandem repeat (STR) testing. Cells were maintained in Dulbecco's Modified Eagle's Medium

(DMEM) supplemented with 10% fetal bovine serum (FBS) and 1% penicillin-streptomycin, and cultured at 37 °C in a humidified 5% $CO_2$ atmosphere.

## Tail vein/intracardiac injection
For the experimental metastasis models, female athymic nude mice were used. For the lung metastasis model, $6 \times 10^6$ cells in 100 μL of PBS were injected per mouse via the tail vein. For the brain metastasis model, a small incision was made in the chest to expose the heart, $2 \times 10^5$ cells in 100 μL of PBS were injected into the left ventricle. To evaluate therapeutic efficacy, mice were treated with either IGUANA-1 (2 mg/kg, administered via intraperitoneal injection) or Silmitasertib (2 mg/kg, administered via oral gavage) once every two days. Metastatic progression was monitored at the indicated time points by

bioluminescence imaging using an Xenogen IVIS system (version 4.0). At the endpoint of the experiment, the mice were euthanized and dissected. The bioluminescence imaging of the extracted mouse brains was performed using a Tanon-5200 Chemiluminescent Imaging System (Tanon). Then the lungs or brains were taken out for 4% PFA fixation and embedded for H&E or IF staining.

For the mouse survival analysis, the observation period was terminated upon the death of the last control group mouse. At this endpoint, the surviving mice in the experimental group exhibited no signs of morbidity, maintaining stable body weight and showing no adverse effects. These mice were right-censored to uphold ethical standards and statistical integrity, as the Kaplan–Meier method appropriately handles censored data[85–87].

### CRISPR screen and data analysis
A custom sgRNA library was designed to target 1,685 metabolic enzyme genes. The sgRNAs for each gene were selected from the human genome-scale CRISPR-Cas9 knockout (GeCKO) library[88]. The sequences of all sgRNAs are provided in Supplementary Data 1. The pooled oligonucleotide library comprising these sgRNAs was synthesized by GENEWIZ.and then cloned into the BsmBI-digested lentiGuide-Puro (Addgene 52963) backbone via Golden Gate assembly. The library was amplified in *E. coli* DH5α with a coverage of >200-fold. Plasmid DNA was prepared via maxi-preparation from the pooled bacteria to generate the final library, which demonstrated >95% sgRNA detection efficiency. The screening experiment was performed as previously described[89]. Briefly, A549 cells were infected with lentivirus expressing Cas9 and blasticidin S deaminase. After one week of selection with blasticidin, single-cell clones were sorted by FACS into 96-well plates. Flag-Cas9 expression in these subclones was confirmed by immunoblotting. The sgRNA library was transduced into A549-Cas9 cells at a multiplicity of infection (MOI) of 0.3. This was followed by puromycin selection for 2 days and subsequent infection with EGFP-encoding virus. On day 5, $6 \times 10^7$ cells were injected into the tail veins of female BALB/c athymic nude mice ($n = 9$). Over 90% of tumor cells were retained in the lung capillaries within 24 h post-injection, consistent with prior studies[19,20]. At the 24-h time point, viable tumor cells were isolated from whole lungs by fluorescence-activated cell sorting (FACS), and sorted cells from every three mice were pooled to form one sample. In parallel, control samples of the input cell population (prior to injection) were also collected. Genomic DNA was extracted from all samples, and the sgRNA inserts were amplified by PCR and subjected to next-generation sequencing, which was performed by Beirui. The CRISPR screening data were analyzed using the MAGeCK algorithm (version 0.5.7) and the MAGeCKFlute R package (version 2.8.0).

### Transwell assay
Transwell migration assays were performed using 6.5 mm diameter Corning BioCoat Control Inserts (8 μm pore size) according to the manufacturer's instructions. Briefly, A549 ($8 \times 10^4$), H1299 ($1 \times 10^5$), and H460 ($1.5 \times 10^5$) cells were harvested post-trypsinization and seeded into the upper chambers in serum-free medium. The lower chambers were filled with medium containing 10% or 20% FBS as a chemoattractant. After incubation for 8 or 18 h, the cells were subjected to methanol fixation and staining with 0.4% crystal violet. The upper side of the membrane was subsequently destained with a cotton swab to remove non-migrated cells. Images of the migrated cells were acquired using an inverted microscope (×200 magnification), and the relative migration was determined by normalizing the total area of the migrated cells to that of the control.

### Microchannel preparation and usage
Polydimethylsiloxane (PDMS, 10/1 w/w PDMS A/crosslinker B) (GE Silicones) was used to prepare 10 μm high micro-channels with

constrictions of varying lengths and 8 μm widths. Prior to loading into the microchannels, cells were stained with DAPI at a 1:10,000 dilution. The cell suspension was then introduced into the channel inlet and allowed to load for 24 h. Subsequently, real-time cell tracking was performed for over 48 h using a Zeiss Celldiscoverer 7 platform, with images captured every 30 min. Cells that died within the microchannels exhibited positive DAPI staining, indicating loss of membrane integrity.

### Wound healing assay
A549 or H1299 cells ($1 \times 10^6$ cells per well) were seeded in 6-well plates. After 12 h, a confluent cell monolayer was scraped using a 200 μL pipette tip. The initial wound width was measured 2 h after scratching, and this time point was designated as 0 h. Images were subsequently captured at 24 or 72 h. The relative cell migration was quantified by measuring the wound width and normalizing it to the width at the 0-h time point.

### Quantitative real-time PCR
Total RNA was extracted using Trizol reagent (Life Technologies), and cDNA was synthesized with the HiScript IV Q RT SuperMix for qPCR kit (Vazyme). Quantitative real-time PCR analysis was performed using a Roche LightCycler384 (version 1.5.1). Data were normalized to expression of a control gene (β-actin) for each experiment. Primer sequences of human ALDH6A1 were 5′-TCAGTGCCAACTGTAAAGCTC-3′(forward) and 5′-TGAGGGACCCGACCAATGA-3′ (reverse); human ALDH9A1 were 5′-GTCGCAGCCGCTCAATTAC-3′(forward) and 5′-CCTTTGCATTTTGAACAGCCAA-3′ (reverse); human ALDH1B1 were 5′-CCCATTCTGAACCCAGACATC-3′(forward) and 5′-AATGACCTCCCCGGTGGTA-3′ (reverse); human PIEZO1 were 5′-GGACTCTCGCTGGTCTACCT-3′(forward) and 5′-GGGCACAATATGCAGGCAGA-3′ (reverse); human ORAI1 were 5′-GACTGGATCGGCCAGAGTTAC-3′ (forward) and 5′-GTCCGGCTGGAGGCTTTAAG-3′ (reverse); human TRPV4 were 5′-GATGGGCGA CCAAATCTGC-3′(forward) and 5′-GAGG ACTCATATAGGGTGGACTC-3′ (reverse); β-actin were 5′-AGAGCTACG AGC TGCCTGAC-3′ (forward) and 5′-AGCACTGTGTT GGCGTACAG-3′ (reverse).

### Cell proliferation assay
A549 cells or H1299 cells were plated in 96-well plates at a density of 1000 cells per well. Following 1, 2, 3, 4, and 5 days of incubation, the plates were harvested, and the cells were fixed using trichloroacetic acid (TCA) and stained with sulforhodamine B (SRB), followed by rinsing of the plates four times with 1% acetic acid and solubilization of the protein-bound dye with 10 mM Tris base solution. The absorbance was measured at 560 nm. Relative cell proliferation was assessed by the OD560 values and normalized to day 1.

### Compression experiment
Flexcell Compression System (Flexcell International Corporation) was utilized to apply the compressive load. The diameters of pulmonary capillaries range from 5 to 8 μm[90]. In microfluidic channels with analogous dimensions (5–8 μm), individual tumor cells experience compressive forces of approximately 80 nN[36]. Using the Flexcell Compression System's standard conversion formula, P (MPa) = (5.65 × Force (lbs))/D (mm²), an applied pressure of 5 kPa in the BioPress well system corresponds to the target cellular force load of 80 nN. A549, H1299, or H460 cells ($1 \times 10^6$) were suspended in 5% GelMA hydrogels (EFL-GM-30), seeded onto glass slides, photo-crosslinked under ultraviolet light, and then subjected to compressive loads of 0 or 5 kPa. Hydrogels was dissolved with GelMA Lysis Buffer (EFL-GM-LS-001), and the cells were collected for immunoblotting and quantitative real-time PCR analysis.

### Stiffness experiment
GelMA hydrogels were used to replicate tissue stiffness. Based on reported lung tissue mechanics (0.5–5 kPa)[48], three GelMA

concentrations were selected to represent distinct ECM stiffness regimes: 10% (~3 kPa, approximating normal lung), 12.5% (~5 kPa, mimicking fibrotic regions), and 15% (~9 kPa, representing pathological stiffness). The hydrogels with different concentrations were added to 6-well plates and crosslinked under ultraviolet light. A549 cells ($5 \times 10^5$ cells/well) were seeded onto the hydrogels in 6-well plates and cultured in growth media for 24 h.

### H&E staining
The brains of the injected mice were removed and fixed with 4% paraformaldehyde (PFA) at 4 °C overnight. Subsequently, the brains were embedded in paraffin. Each sample was completely sectioned at 50 µm intervals with a thickness of 5 µm. For histological examination, standard hematoxylin and eosin (H&E) staining was carried out. The images of H&E staining were taken and analysed using ImageJ software.

### Immunofluorescence/TUNEL staining
For the detection of tumor cell death, lungs were harvested from the injected mice at 24 h post-injection. For the analysis of tumor cell extravasation, lungs were harvested at 48 h post-injection. All harvested lungs were fixed overnight at 4 °C with 4% PFA. After cryosectioning, sequential 10 µm lung tissue sections were prepared at 50 µm intervals for analysis. Immunofluorescence analysis was carried out with an anti-CD31 antibody (diluted 1:100) to enable histological examination of the vasculature. Concurrently, the DeadEnd Fluorometric TUNEL System was applied according to the manufacturer's guidelines to detect tumor cell death in the same sections.

The expression of ALDH1B1, 4-HNE, and RelA in tumor cells from both mouse and human lung tissues was evaluated using a standard immunofluorescence protocol. Briefly, tissue sections (either frozen from mice or paraffin-embedded from humans) were fixed and incubated with specific primary antibodies (1:200 dilution), followed by corresponding Alexa Fluor-conjugated secondary antibodies (1:1000) and DAPI for nuclear counterstaining. Images were acquired using an Olympus FV3000 confocal microscope. Protein expression intensity within the confining capillaries was semi-quantitatively scored by analyzing three to five randomly selected fields (at 400× magnification) per section, and the mean score was calculated for each sample.

Semiquantitative scoring of intensity was calculated using Columbus (PerkinElmer) as described previously[91]. Briefly, immunofluorescence images were analyzed using Columbus software (version 2.8.2). Cell numbers were quantified by nuclear identification through DAPI staining (blue). Capillary regions (confining vs. nonconfining) were demarcated using CD31 staining (white). ALDH1B1 and 4-HNE intensities were measured within these regions. Nuclear RelA intensity was quantified within DAPI-defined nuclei in these regions.

### Immunoprecipitation and immunoblotting analysis
For co-immunoprecipitation (co-IP) assays, a mild co-IP buffer (50 mM Tris-HCl pH 7.5, 1% Triton ×-100, 0.01% SDS, 150 mM NaCl, 1 mM dithiothreitol, 0.5 mM EDTA, 0.1 mM sodium orthovanadate, 0.1 mM sodium pyrophosphate, 0.1 mM sodium fluoride, and protease inhibitor cocktail) was used to preserve native protein-protein interactions. For enzyme activity or in vitro kinase assays, a more stringent IP buffer was used, which contained 0.1% SDS was used, with all other components identical to the co-IP buffer. Flag-tagged proteins were immunoprecipitated using anti-Flag M2 affinity agarose.

For immunoblotting, cells were lysed in cold RIPA buffer supplemented with a 1× protease inhibitor cocktail for 30 min on ice, with vortexing at 10-min intervals. The lysates were centrifuged at $14,000 \times g$ for 10 min at 4 °C to collect the supernatant. These protein extracts were combined with loading buffer, heated at 95 °C for 10 min to denature the proteins, and then subjected to SDS-PAGE. The separated proteins were electrophoretically transferred onto nitrocellulose membranes. For immunodetection, the membranes were first incubated with specific primary antibodies at 4 °C overnight. The antibodies and their dilutions were as follows: ALDH1B1 (1:4000), CSK23 (1:1000), Flag (1:4000), HA (1:4000), Actin (1:4000), Tubulin (1:5000), RelA (1:2000), IKKβ (1:1000), IκBα pS32/36 (1:1000), IKKβ pS177/181 (1:1000), IκBα (1:4000), Lamin B1 (1:4000). Subsequently, the membranes were incubated with HRP-conjugated secondary antibodies (diluted 1:5000) for 1 h at room temperature. After thorough washing with TBST, immunoreactive bands were visualized using enhanced chemiluminescence (ECL)[92].

### Subcellular fractionation analyses
Cytoplasmic and nuclear proteins were extracted from treated cells using a modified protocol as described previously[93]. Briefly, cells were incubated with a plasma membrane lysis buffer (200 mM HEPES pH 7.9, 200 mM KCl, 25 mM MgCl$_2$, 6 M sucrose, 50% glycerol, 0.5% NP-0.4, 0.5 mM DTT, and protease inhibitor cocktail) at 4 °C for 20 min to lyse the plasma membranes. The lysates were centrifuged at $1000 \times g$ for 5 min at 4 °C, and the supernatant was collected as the cytoplasmic fraction. The nuclear pellet was then resuspended in PBS and washed three times ($1000 \times g$, 5 min, 4 °C) to remove cytoplasmic contaminants. Finally, the purified nuclei were resuspended in 50 µL of cold RIPA buffer supplemented with 1× protease inhibitor cocktail and lysed overnight at 4 °C with agitation to obtain the nuclear fraction. The extracted proteins from both fractions were then subjected to immunoblotting with the corresponding antibodies.

### Purification of recombinant proteins
GST-IKKβ and HIS-CSK23 were expressed in bacteria and purified. Briefly, the vectors expressing GST-IKKβ and HIS-CSK23 were used to transform BL21/DE3 bacteria. Then 0.5 mM isopropyl-beta-D-thiogalactopyranoside was used to induce protein expression (20 h, 16 °C). Cell pellets were collected and sonicated in PBS with the addition of proteasome inhibitors before centrifugation at $12,000 \times g$ for 60 min (4 °C). Cleared lysates were then bound to glutathione resin (Genescript) or Ni-NTA resin (Genescript) with rolling (3 h, 4 °C). Beads were washed extensively before eluting for 1 h in GST elution buffer (50 mM Tris-HCl, 10 mg ml$^{-1}$ glutathione, 300 mM NaCl, pH 8.0) or HIS elution buffer (PBS (pH 7.4) plus 500 mM imidazole). Eluted proteins were then dialysed against PBS.

### In vitro kinase assay
The kinase reactions were performed as described previously[94]. In brief, GST-IKKβ and HIS-CSK23 were purified from bacterial. SFB-IKKβ and Flag-CSK23 were immunoprecipitated from A549 cells. The IKKβ and CSK23 proteins were mixed with 1× kinase reaction buffer (50 mM HEPES (pH 7.5), 1 mM EGTA, 0.01 mM BSA, 0.01 % Tween-20, 10 mM MgCl$_2$, 2 mM MnCl$_2$, 2 mM DTT) in a 50-µL reaction containing or not containing 100 µM ATP. The samples were then incubated at 30 °C for 1 h and stopped by adding 4× SDS loading buffer. Phosphorylated IKKβ protein was detected by immunoblotting.

### ALDH1B1 activity assay
The enzymatic activity of ALDH1B1 was measured as described previously[13]. In brief, the Flag-ALDH1B1 and Flag-ALDH1B1 E285A proteins were immunoprecipitated from A549 cells. The reaction was initiated by adding the acetaldehyde into the reaction mixture containing 0.1 M sodium pyrophosphate, 1 mM NAD$^+$, 1 mM pyrazole, 1 mM 2-mercaptoethanol, and ALDH1B1 WT/E285A protein, then monitoring the formation of NADH at 340 nm for 25 min at 25 °C.

### In situ cell death detection
After 6 h of culture in transwell inserts ($6 \times 10^4$ cells), the cells were stained using an Annexin V-PE kit according to the manufacturer's instructions. Images were acquired and analyzed using a BioTek Cytation 5 imaging system (version 3.15). Transwell pore-confined

cells and non-pore cells were distinguished and selected based on bright-field imaging. The cell death percentage was calculated as the ratio of Annexin V-PE-positive cells (either pore-confined or non-pore) to the total number of cells counted for each population.

## Lipid peroxidation measurement

After 6 h of culture in transwell inserts ($6 \times 10^4$ cells/insert), the cells were stained with 0.5 μM BODIPY 581/591 C11 for 30 min at 37 °C to assess lipid peroxidation. This ratiometric fluorescent probe quantitatively detects ferroptosis-associated lipid peroxidation. In its reduced state, the probe emits red fluorescence (excitation/emission: 581/591 nm), which shifts to green fluorescence (ex/em: 488/510 nm) upon oxidation by lipid hydroperoxides. The relative lipid peroxidation levels in pore-confined and non-pore cells were determined by calculating the intensity ratio of oxidized (green, F510) to reduced (red, F591) fluorescence signals (F510/F591), measured using a PerkinElmer Operetta high-content imaging system. Data analysis was performed using Harmony software.

## GSH/GSSG measurement

To measure intracellular GSH and GSSG levels, A549 or H1299 cells ($1 \times 10^6$) were seeded in serum-free medium into the upper chamber of a 24 mm diameter Transwell (Corning BioCoat Control Insert, no ECM, 8 μm pore size). The lower chamber was filled with medium containing 20% FBS as a chemoattractant. After 18 h of incubation, those cells remaining on the upper surface of the membrane (non-migratory) or within its pores (confined) were collected. The intracellular concentrations of GSH and GSSG in these cells were quantified using a commercial assay kit according to the manufacturer's instructions, with values derived from a standard curve.

## Caspase-3/7 activity assay

For the caspase-3/7 activity assay, A549 cells ($1 \times 10^6$) were seeded into a 24 mm diameter transwell chamber in serum-free medium. After 18 h, the chamber membrane was treated with trypsin, and the cells adherent to the upper surface of the membrane and confined in the pore were collected. Caspase-3/7 activity was measured using an assay kit following the manufacturer's protocol, with quantification based on a standard curve.

## GEO data analysis

Deep sequencing data from this study are publicly available in the GEO database (http://www.ncbi.nlm.nih.gov/geo) (GSE72094 and GSE37745). The RNA-seq sample expression levels were converted from fragments per kilobase of transcript per million mapped reads (FPKM) to transcripts per million (TPM), and then $\log_2(\text{TPM} + 1)$ was calculated. Differences in prognostic outcomes were assessed using Kaplan–Meier analysis in R software with the "survival" and "survminer" packages[95]. All statistical analyses were carried out using R software (version 4.3.2).

## Luciferase reporter assay

Transcriptional activation of ALDH1B1 and NF-κB in A549 or H1299 cells was measured with a luciferase reporter assay as described previously[96]. In brief, cells were co-transfected with a firefly luciferase reporter plasmid under the target promoter and a Renilla luciferase control plasmid. Using the Promega Dual-Luciferase Reporter Assay Kit, relative luciferase activity was normalized to that of untreated cells, empty vector-expressing cells, or the Renilla control.

## Chromatin immunoprecipitation

ChIP assays were performed as previously described[97]. Briefly, chromatin complexes were immunoprecipitated using an anti-RelA rabbit monoclonal antibody, with rabbit monoclonal IgG serving as a negative control. Recruitment of RelA to the ALDH1B1 promoter region was quantified by real-time PCR, using 2% of the pre-immunoprecipitation, sonicated chromatin as the input control. Primer sequences used for the amplification were 5'-TTCGGTCTCCGAAAACTTTGCTG-3'(forward) and 5' -AGGCTCCTAAGGCACTGGTTT-3'(reverse).

## Mass spectrometry analysis

Co-immunoprecipitation and Protein Separation. SFB-IKKβ protein complexes were immunoprecipitated from A549 cells subjected to 0 kPa or 5 kPa pressure stimulation. The precipitated complexes were denatured by heating at 95 °C for 10 min, followed by electrophoretic separation of IKKβ-associated proteins using SDS-PAGE.

Protein Digestion and Peptide Preparation. Proteins were precipitated from excised gel bands using cold acetone. The resulting protein pellets were resuspended in 8 M urea/100 mM Tris-HCl (pH 8.5) buffer. Reduction was performed with 5 mM TCEP (Thermo Scientific) for 30 min at room temperature, followed by alkylation with 10 mM iodoacetamide (Sigma) for 20 min. The protein solution was then diluted fourfold and digested overnight with trypsin at a 1:50 (w/w) enzyme-to-substrate ratio. The digestion was terminated by acidification with formic acid, and resulting peptides were desalted using Monospin C18 columns (SHIMADZU-GL).

LC-MS/MS Analysis. Desalted peptides were separated on a custom-packed 30 cm analytical column (75 μm ID, ReproSil-Pur C18-AQ 1.9 μm resin, Dr. Maisch GmbH) maintained at 55 °C using an EasynLC 1200 system (Thermo Scientific). The mobile phase consisted of 0.1% formic acid in water (buffer A) and 0.1% formic acid in 80% acetonitrile (buffer B). Peptides were eluted with a multi-step gradient: 2–10% B over 1 min, 10–35% B over 80 min, 35–60% B over 15 min, 60–100% B over 15 min, and held at 100% B for 9 min, at a constant flow rate of 300 nL/min.

Data Acquisition and Processing. MS data were acquired on a Q Exactive Orbitrap mass spectrometer (Thermo Scientific) operating in data-dependent mode. Peptides were ionized using a 2.5 kV spray voltage. Full MS scans (m/z 300–1800) were acquired at 70,000 resolution, followed by fragmentation of the top 20 most intense precursors at 17,500 resolution. Automatic gain control targets were set at 3e6 for MS and 5e5 for MS/MS scans, with maximum injection times of 50 ms and 100 ms, respectively. Dynamic exclusion was set to 15 s.

MS/MS data were processed using MaxQuant (v1.6.10.43) against the human UniProtKB database. Search parameters included: tryptic digestion with up to two missed cleavages; carbamidomethylation (C) as fixed modification; oxidation (M) and protein N-terminal acetylation as variable modifications; precursor mass tolerance of 20 ppm. Protein quantification was performed using the LFQ intensity algorithm with a minimum of six amino acids required for peptide identification.

## Cell aldehyde measurement

For aldehyde measurement, A549 or H1299 ($1 \times 10^6$) cells were collected after trypsinization and added to a transwell chamber with a 24 mm diameter, containing serum-free medium. The outside the chamber was filled with medium containing 20% FBS. After 18 h, the membrane of the chamber was immersed in trypsin to collect the cells adherent to the upper surface of the membrane and the cells confined within the pores. These cells were then sonicated for 7 min, followed by centrifugation at 3500 × g for 15 min at 4 °C. The supernatant was transferred to a 1.5 mL Eppendorf tube, and 5 mM DNPH solution (1:20) was added. The mixture was incubated in the dark at 37 °C with constant agitation for 1 h. After incubation, the mixture was centrifuged at 1000 × g for 5 min, and 4 mL of hexane was added. The mixture was vortexed and then centrifuged at 4000 × g for 5 min. The upper layer

containing hexane was collected, and the sample was subsequently evaporated at 4 °C.

Samples were reconstituted in 80% methanol solution prior to mass spectrometric analysis. Chromatographic separation was performed on a Waters ACQUITY UPLC HSS T3 column (100 mm × 2.1 mm, 1.8 μm) maintained at 40 °C, using a mobile phase comprising 0.1% (v/v) formic acid in water (solvent A) and 0.1% (v/v) formic acid in methanol (solvent B). The elution protocol employed a gradient program: 10% B (0–1 min), 10–95% B (1–5 min), 95% B (5–8 min), 95–10% B (8–8.1 min), and 10% B (8.1–12 min), with a constant flow rate of 0.3 mL/min and 1 μL injection volume.

Mass spectrometric detection was carried out using an Agilent 6495 triple quadrupole instrument coupled to an Agilent 1290 Infinity II UHPLC system. Analysis was performed in negative ionization mode with the following optimized parameters: gas temperature 200 °C; drying gas flow 14 L/min; nebulizer pressure 20 psi; sheath gas temperature 250 °C; sheath gas flow 11 L/min; capillary voltage 3000 V; and nozzle voltage 1500 V. Aldehydes were quantified in multiple reaction monitoring (MRM) mode, with specific transition pairs detailed in Supplementary Data 4. All data acquisition and processing were performed using Agilent QQQ Quantitative Analysis Software (v10.2).

### Statistics and reproducibility

No statistical methods were used to predetermine sample size, but our sample sizes are similar to those reported in previous publications[16,22]. All statistics were performed using GraphPad Prism software (version 8.0). Specific statistical tests for each experiment are described in the figure legends. A value of $P < 0.05$ was considered statistically significant.

### Reporting summary

Further information on research design is available in the Nature Portfolio Reporting Summary linked to this article.

## Data availability

Source data are provided with this paper. The mass spectrometry proteomics data have been deposited in the ProteomeXchange Consortium (https://proteomecentral.proteomexchange.org/cgi/GetDataset?ID=PXD061376) via the iProX partner repository with accession code PXD061376 (https://www.iprox.cn/page/project.html?id=IPX0011233000). Deep sequencing data are available in the GEO database under accession codes GSE72094 and GSE37745. This paper does not report original code. All the other data are available within the article and its Supplementary Information. Source data are provided with this paper.

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

## Acknowledgements

This work was supported by the National Key R&D Program of China (2022YFA0806200 and 2024YFA1306003) to W.Y.; The National Natural Science Foundation of China (32521007, 92357301, 92253305 and 32025013) to W.Y.; The Strategic Priority Research Program of the Chinese Academy of Science (XDB0990000) to W.Y.; Science and Technology Commission of Shanghai Municipality (24J12800600); the Research Funds of Hangzhou Institute for Advanced Study, UCAS (2025HIAS-ZL014); CAS Project for Young Scientists in Basic Research (YSBR-014); Shanghai Municipal Science and Technology Major Project; W.Y. is a SANS Exploration Scholar. The National Natural Science Foundation of China (32200625) to M. Liu; The National Natural Science Foundation of China (8237112887) to F. Yao; Shanghai Sailing Program (22YF1453900) to M. Liu; The Shanghai Oriental Talent Program to Y. Zhang. We thank Yanjun Liu (Fudan University, Shanghai, China) for providing PDMS on the manuscript. We thank Bo Jiang (SIBCB, CAS, Shanghai, China) for his support with the animal experiments. We thank Jinhong Li (SIBCB, CAS, Shanghai, China) for his assistance in analyzing the CRISPR-Cas9 screening data. We thank the Genome Tagging Project (GTP) Center and the Core Facilities of CEMCS for technical support. We thank Shuai Han and Hongwei Zhao of the Chemical Biology Core Facility in CEMCS for technical support. We thank Chen Su of the Mass Spectrometry System at the National Facility for Protein Science in Shanghai (NFPS). We thank Xiaoyan Xu and Mass Spectrometry & Metabolomics Core Facility of Westlake University for LC-MS/MS analysis.

## Author contributions

W.Y. conceived this study. W.Y. and B. Liu designed the study. B. Liu performed the majority of experiments, conducted data analysis, and contributed to figure editing. M.L. provided experimental assistance and participated in data analysis. Y.Z. offered technical support and constructive suggestions. Y.Zhu. and D.Z. analyzed TCGA data and CRISPR-Cas9 screening results. H.G. assisted in manuscript review. F.Y., D.G., and Y.Zhao provided experimental support. B.T. and F.Yao. contributed reagents and pathological. W.Y. wrote the manuscript with comments from all authors.

## Competing interests

The authors declare no competing interests.
