## [Transparent Peer Review file · Nature Communications]

Compression-induced NF- κ B activation sustains tumor cell survival in confinement by detoxifying aldehydes and promotes metastasis

Corresponding Author: Dr Weiwei Yang

This file contains all reviewer reports in order by version, followed by all author rebuttals in order by version

Attachments originally included by the reviewers as part of their assessment can be found at the end of this file..

Version 0:

Reviewer comments:

Reviewer #1

(Remarks to the Author)

In this study, Liu et al. explore how cancer cells adapt metabolically to migrate through confined spaces, a key step in metastasis. Researchers conducted an in vivo CRISPR knockout screen targeting 1,684 metabolic enzymes in a lung metastasis mouse model. They identified ALDH1B1 as crucial for tumor cell survival within capillaries. Under compressive forces, CSK23 phosphorylates IKK β , activating the NF- κ B pathway, which increases ALDH1B1 expression. This upregulation enhances aldehyde detoxification, suppresses ferroptosis, and aids cell survival during capillary migration. Inhibiting CSK23 or ALDH1B1 impairs metastasis, highlighting them as potential therapeutic targets. Elevated ALDH1B1 and NF- κ B activation in patient tumor tissues correlate with metastatic recurrence, underscoring their role in metastasis. While the association of ALDH1B1 upregulation with tumor metastasis and survival has been previously established, the authors have made significant strides in elucidating the mechanism by which tumors adapt to compressive forces within confined 3D spaces such as capillaries. With moderate revisions to the text and figures, this manuscript has the potential to be a strong fit for publication in a rigorous journal like Nature Communications.

Major concerns:

1. Authors should explain the screen setup in more details. For example, when dissecting mouse lung tissue, how to differentiate cancer cell in the alveoli or in the lung capillary tract? Also, authors need to show some basic quality control metric of the screen, at least in the supplementary to demonstrate the coverage and selection power of the screen.
2. PIEZO1 and its sensing of the compression force were shown to affecting the expression level of ALDH1B1. Does depletion of PIEZO1 therefore also affecting the survival of the tumor cells in tight capillaries? It seems this part of data is missing from the Figure 3.
3. Liu et al. have elucidated the signaling pathway from compression force sensing to upregulation of ALDH1B1. However, there remain one missing piece of the puzzle: how PIEZO1 activate CSK23? PIEZO1 is a calcium channel that activated by force, could there be a calcium sensing messenger between PIEZO1 and CSK23?
4. In Figure 6d, without any CSK23 depletion, IKKbeta pS177/181 level is lower in compression 1hr condition than the compression 0.5hr condition, which is contradicting with result in Figure 5b. What causes this discrepancy? The authors should quantify their western blot results and show statistical replicates.

Minor concerns:

1. The authors should introduce briefly (a sentence is sufficient) on the background of A549 cell line, which will help the reader understand rationale of using it.
2. Figure 2a, right panel. It would be interesting to see the statistical comparison between group 1 and 3 (i.e., negative control vs shALDH1B1+ALDH1B1 rescue). It seems rescue on has even lower cell death rate.
3. Please explain the assay briefly for Figure 2f. For example, what's the green/red ratio. It would be easier to understand for the audience that are not familiar to ferroptosis.
4. Although treatment is not the focus of the study, the authors should attempt to discuss about the therapeutic potential (pros and cons) of ALDH1B1 inhibitor (or any method perturbing this cancer adaption pathway).

Reviewer #2

(Remarks to the Author)

This manuscript presents a CRISPR knockout screen targeting 1,684 metabolic enzymes in a mouse model of lung cancer metastasis. The authors identify ALDH1B1, a member of the aldehyde dehydrogenase family, as essential for tumor cell survival within physically confining capillaries. Mechanistically, they propose that compressive forces activate casein kinase 2A3 (CSNK2A3), which in turn phosphorylates IKK β (I κ B kinase subunit beta) at serine residues 177 and 181. This phosphorylation event is suggested to activate the NF- κ B pathway, leading to upregulation of ALDH1B1 expression. The increased expression of ALDH1B1 is proposed to enhance aldehyde detoxification, suppressing ferroptosis and supporting tumor cell survival in confined environments, ultimately facilitating lung cancer metastasis. They further suggest that genetic or pharmacological inhibition of CSK23 or ALDH1B1 could impair lung cancer metastasis.

The central novelty and impact of the study lies in linking mechanical compression within capillaries to changes in ALDH isoform expression and promoting metastasis via ferroptosis suppression. The signaling work is of high quality and rigor, and the graphical summary at the end of the manuscript is especially well-done. However, despite the novelty, there are several significant concerns that must be addressed before publication:

Concerns

1. These was limited scope of ALDH isoform screening. Specifically, the ALDH family includes 18 isoforms, many of which are implicated in cancer progression and metastasis. It is unclear whether all ALDH isoforms were included in the gRNA screen targeting 1,684 metabolic enzymes. Since isoform redundancy is well-documented—where suppression of one ALDH isoform results in compensatory upregulation of another—focusing solely on ALDH1B1 is insufficient. The manuscript must clarify whether other ALDH isoforms were also screened and ruled out. Controls targeting other ALDH isoforms (e.g., ALDH1A3, ALDH2, ALDH3A1) should be included to confirm the specificity of the phenotype.
2. There are limitations of the experimental metastasis assay that are not addressed. Specifically, in experimental metastasis assays, many cancer cells accumulate in the lungs initially, but the majority are naturally cleared within 24 hours. The authors need to clarify whether the genes identified are truly mediating survival in the confined capillaries or merely affecting initial accumulation. A clear distinction between transient lodging and sustained survival in capillaries must be made.
3. Figures as presented suggest cell line limitation and lack of generalizability. Specifically, the study relies almost entirely on the A549 lung cancer cell line. To demonstrate robustness and reproducibility, results must be validated across additional lung cancer cell line and ideally across different cancer types. Partial validation of some experimentation in only the H1299 cell line is not sufficient. Figures throughout the manuscript should be updated to include these validations.
4. There is concern related to the scope of confinement in the metastatic cascade. Specifically, the metastatic process involves several stages of physical confinement—not just within capillaries but also during tissue migration, intravasation, circulation, and extravasation. The manuscript currently focuses only on capillary confinement. The authors must experimentally address whether the described mechanism applies more broadly to other stages of the metastatic cascade.
5. There is major concern regarding the possible redundancy among ALDH isoforms in metastasis. Specifically, given the overlap in detoxification function across ALDH isoforms, it's likely that multiple isoforms could suppress ferroptosis under confined conditions. The manuscript must explore and experimentally demonstrate whether other ALDH isoforms are also upregulated or functionally active in response to mechanical stress and assess their potential contributions.
6. The therapeutic implications are overstated in several parts of the manuscript. Specifically, the suggestion that ALDH1B1 could be a promising therapeutic target is problematic. Many prior efforts to target specific ALDH isoforms have failed due to functional compensation by other isoforms. A more realistic approach would be to explore broad-spectrum ALDH inhibitors, which is not currently tested or discussed in the manuscript.
7. Calibration of the in vitro model validation for mechanical compression is a concern. Specifically, the BioPress well system used to model mechanical compression must be better validated. The authors need to demonstrate that the in vitro forces and extracellular matrix (ECM) conditions are comparable to those experienced by tumor cells in capillaries or at other sites in the metastatic cascade. The contribution of ECM stiffness to gene expression changes should also be evaluated since it is significant in the scientific literature.
8. Western blot quality and quantification needs improvement. Specifically, several western blot images appear overexposed for the loading controls, which may not be within the linear detection range. The authors should provide lighter exposures and include quantitative analysis normalized to loading controls.
9. There is weakness of human tumor correlation data (Figure 7). Specifically, the correlative data from human tumors presented in Figure 7 are interesting but lack depth. Stronger validation using larger datasets or orthogonal approaches (e.g., IHC, RNA-seq, or survival analysis) would improve the translational relevance of this finding.

Reviewer #3

(Remarks to the Author)

The paper from Liu et al (NCOMMS) proposes that compression force in capillaries activates NF- κ B during metastasis to promote survival of tumor cells through upregulation of ALDH1B1. The authors have used metastasis models (tail vein injection and intra-cardiac injection) to measure metastasis and analyze tumor cells in capillaries. In vitro studies, using confined pore transwell assays indicate that CSK23 is activated to phosphorylate IKK to promote NF- κ B activation which drives expression of ALDH1B1. Mutant ALDH1B1 is inactive in promoting the confinement-induced cell survival response, consistent with the effect of ALDH1B1 on acetaldehydes.

While the cell-based studies are consistent with the hypothesis, there is a concern with the tumor metastasis studies. Human lung cancer patient samples were analyzed to correlate with animal and cell-based studies. The authors have failed to acknowledge the extensive publication history on ALDH1B1 in cancer (over a 100 publications), where it is described to promote a variety of oncogenic phenotypes such as control of cancer stem cells and beta-catenin signaling (both involved in metastasis). Lack of consideration of these oncogenic phenotypes confounds interpretation of some of the proposed results. Additional concerns are outlined below.

- 1) In Fig. 1, the authors conclude that loss of ALDH1B1 in tumors promotes metastasis via effects on survival of tumor cells in capillaries. Given the significant role of ALDH1B1 in several cancer phenotypes (unreferenced, and see statement above), the authors cannot conclude that the loss on metastasis is solely through cell death in capillaries. In this regard, can the authors show that the same number of cells reach the lung for WT vs ALDH1B1 knockdown? Is that what is meant to be shown in fig. 1g? The authors need to analyze the number of mets that form and not total GFP intensity since loss of ALDH1B1 will likely reduce the growth of the metastases. The authors utilize TCGA data to indicate that elevated levels of ALDH1B1 mRNA is linked with metastasis and survival.
- 2) Related to the point above, the authors use Kaplan-Meier human data to argue that ALDH1B1 promotes metastasis. In the cases with higher ALDH1B1, it could be that the primary tumor is larger, has more cancer stem cells, or is resistant to therapy. Additionally, what is measured in the Kaplan-Meier analysis is ALDH1B1 RNA expressed in the tumor and not in tissue capillaries, thus making this point disconnected from the proposed mechanism of upregulation in confined capillaries. The lack of consideration of additional oncogenic phenotypes for ALDH1B1 is a significant flaw in the manuscript.
3. For the studies shown in Fig. 2 related to survival and other effects on confined transwell pore assays, the authors need to show side by side analysis with unconfined cells. In the cell studies related to suppression of ferroptosis by ALDH1B1, did the authors analyze apoptosis? (this relates to a point below).
4. For the metastasis studies using the ALDH1B1 inhibitor, again the authors need to show effects on metastasis numbers and size, and not luciferase intensity (this is focused on the concept that inhibiting ALDH1b1 will likely block growth of mets – unrelated to effects on actually promoting metastasis).
5. Fig. 3 shows that the mechanosensory protein PIEZO1 is required for activation of ALDH1B1. Effects are called “dramatic” but are 2-fold or less.
6. PDTCC blocked the upregulation of ALDH1B1, and the authors took to mean that the IKK/NF- κ B pathway was involved. Note that PDTCC can block a number of pathways. They authors note that levels of RelA are increased in compression/confinement but this experiment does not measure nuclear RelA. Knockdown of RelA blocked compression/confinement-induced ALDH1B1. The authors linked confinement induced cell death with ferroptosis (and see comment below). As before, confinement results with non-confinement results need to be placed together.
7. Compression induces the phosphorylation of IKKbeta, and this correlated with p-IkBa (in the figures the authors need to refer to this correctly – IkappaB-alpha - with Greek letters). In Fig 5d, the effect on an NF- κ B reporter is relatively minimal. Knockdown of IKKbeta was shown to reduce ALDH1B1 levels.
8. The authors found that CSK3 associates with IKKbeta during tumor cell compression and promotes IKKbeta activity, Knockdown of CSK3 leads to loss of ALDH1B1 expression and promoted cell death.
9. In Fig. 6, does knockdown of CSK3 and the CSK3 inhibitor block mets (number and size) – see points above. Like ALDH1B1, CSK3 is linked with oncogenic phenotypes – not referenced. The question is whether loss of CSK3 truly affecting metastasis or simply growth of the tumors once they have metastasized.
10. Fig. 7 links ALDH1 levels with capillary tumor cells and metastasis in lung cancer patients. These experiments are examining migration of cells from the tumor site to relate to distal metastasis and survival. There are several questions/concerns. When the authors refer to unconstrained tumor cells, are they simply referring to tumor? The cell death measurement is TUNEL assays which is associated with apoptosis and less with ferroptosis (see point above – did they authors measure apoptosis in their cell-based studies? Lastly, the text says that the authors are measuring nuclear RelA, but there is nothing to describe how that was measured. It seems that they are staining for whole cell RelA – but only nuclear would be meaningful for the overall hypothesis.

Reviewer #4

(Remarks to the Author)

The manuscript by Liu et al. investigates the role of ALDH1B1 in promoting lung cancer metastasis by enhancing cell survival and motility. The authors suggest that ALDH1B1 is regulated by the CSK23-RelA/NF- κ B pathway and propose Piezo1 as a mechanosensor that initiates this signaling pathway in lung cancer.

The use of a CRISPR knockout screening system to identify key metabolic enzymes is a sophisticated and logically sound approach. However, the method used to identify Piezo1, specifically through the application of a tumor microenvironment-mimicking platform, is not appropriate for lung cancer. Given the presence of numerous other mechanosensitive calcium channels, such as those in the Orai or TRPV families, the authors have not sufficiently explored alternative possibilities for mechanosensing in lung cancer metastasis. Consequently, the claim that Piezo1 is the primary mechanosensor driving CSK23-RelA signaling lacks a strong logical foundation.

Additionally, it remains unclear whether aldehyde levels are elevated in the confined space of the lung. The authors should demonstrate a clear relationship between aldehyde production and the metastatic progression of lung cancer to support their

hypothesis.

Furthermore, the study reports brain metastasis occurring within seven days after lung cancer inoculation into the lung. Is there any existing literature supporting such a rapid metastatic timeline? The authors should provide references or additional experimental validation for this observation.

The definition and generation of confined space in this study also require further clarification. While lung cancer cells may experience mechanical constraints within capillaries during metastasis, anoikis or pulsatile hydrostatic pressure could play a more significant role than simple confinement. The authors have not considered the possibility that different types of mechanical stress might influence lung cancer behavior. A more comprehensive discussion of these factors is necessary to strengthen the study's conclusions.

Overall, while the study presents an interesting perspective on ALDH1B1-mediated metastasis, the manuscript requires a more logically structured approach, further experimental validation, and consideration of alternative mechanosensors and mechanical stress factors.

Reviewer #5

(Remarks to the Author)

In this manuscript, Liu et al. describe metabolic adaptations that enable cancer cells to survive and metastasize within confining microenvironment. Using an in vivo CRISPR-based knockout screening system, the authors identify ALDH1B1 as a critical target for tumor cell survival within capillaries. Mechanistically, they demonstrate that compressive forces activate CSK23, which phosphorylates IKK β at S177/181, triggering NF- κ B-dependent upregulation of ALDH1B1. Elevated ALDH1B1 enhances aldehyde detoxification, suppresses ferroptosis, and promotes tumor cell survival, highlighting CSK23/ALDH1B1 as potential therapeutic targets for metastatic cancer. Overall, this work uncovers a connection between mechanical stress, NF- κ B signaling, and aldehyde metabolism during metastatic progression. Addressing the major concerns would strengthen the manuscript, the detail review is outlined below:

Major points and suggestions :

1. The authors used tail vein injection of gene-edited A549 cells to observe genetic differences in cells within pulmonary capillaries. How did the authors exclude the possibility that gene knockout affected cell proliferation, leading to false-positive results of reduced vascular cell counts? Additionally, is this model original? Is there any other literature that uses this method?
2. The authors proposed that confined cells sense mechanical stress through Piezo1 and ultimately express ALDH1B1 to mitigate aldehyde toxicity. Why is this pathway specific to confined cancer cells? Why do normally growing cells not require similar mechanisms to maintain growth? Is it because confined cells face higher ferroptotic stress? Please provide experimental data to clarify.
3. In Patient 2's results (not enlarged), ALDH1B1 expression showed poor co-expression with CD31, and high ALDH1B1 expression was apparently observed in non-confined regions. However, the statistical plot in Fig. 7b does not reflect this trend. The authors should check the representative results, carefully verify the statistical methods, and accurately present the statistical outcomes.

Minor comments:

1. Western blot images in high-quality journals should include molecular weight markers.
2. No information on tumor patient tissues or ethical approval is provided in the manuscript.
3. Please confirm the appropriateness of using the GEO database (GSE72094), which includes tumors with Kras and other genes mutations.
4. In Fig. 1h, TUNEL-positive areas do not fully overlap with ALDH1B1-knockdown cells, and non-tumor dead cells may affect the result. The authors should re-select representative images or quantify the percentage of co-expression between TUNEL-positive and mCherry-positive regions.
5. The method section should highlight the sampling of pulmonary capillaries.
6. Please carefully verify all results for data integrity (e.g., no endpoint data for the experimental group in Fig. 2m).
7. Nuclear and cytoplasmic fractions in Fig. 5c should be presented on the same gel to compare.
8. Total protein loading controls are required for phosphoprotein detection in Fig. 6d.
9. Supplementary images are excessively compressed, compromising readability.

Reviewer #6

(Remarks to the Author)

Reviewer #7

(Remarks to the Author)

Version 1:

Reviewer comments:

Reviewer #3

(Remarks to the Author)

The authors have made a thorough effort at responding to the critique. Their responses address have satisfactorily addressed my concerns.

Reviewer #4

(Remarks to the Author)

In this manuscript, the authors identify a role of ALDH1B1 regulated by CSK23-RelA/NF-kB and Piezo1 during lung cancer metastasis. Overall, the authors have performed a thorough and comprehensive study supporting their hypothesis that CSK23 and ALDH1B1 play an important role in metastatic environment of lung cancer. Over the past several years, studies have highlighted the critical role of biomechanical forces in cancer metastasis. In this context, this work is an important study that elucidates how the compression forces experienced within extremely narrow blood vessels in lung cancer specific metastasis niches influence lung cancer dissemination and proliferation, as well as the underlying mechanisms. In the revised manuscript, the authors have included additional control data that address my concerns and have strengthened the manuscript.

Reviewer #5

(Remarks to the Author)

This is an excellent piece of work. The authors have provided a thorough and in-depth mechanistic study, and their well-designed experiments have fully addressed the concerns I raised previously.

Reviewer #6

(Remarks to the Author)

Reviewer #7

(Remarks to the Author)

Reviewer #8

(Remarks to the Author)

The comments by reviewer #2 were satisfactorily addressed by the authors, and the manuscript as revised is appropriate for publication.

REVIEWER COMMENTS

Reviewer #1 (Remarks to the Author):

Reviewer Summary:

In this study, Liu et al. explore how cancer cells adapt metabolically to migrate through confined spaces, a key step in metastasis. Researchers conducted an in vivo CRISPR knockout screen targeting 1,684 metabolic enzymes in a lung metastasis mouse model. They identified ALDH1B1 as crucial for tumor cell survival within capillaries. Under compressive forces, CSK23 phosphorylates IKK β , activating the NF- κ B pathway, which increases ALDH1B1 expression. This upregulation enhances aldehyde detoxification, suppresses ferroptosis, and aids cell survival during capillary migration. Inhibiting CSK23 or ALDH1B1 impairs metastasis, highlighting them as potential therapeutic targets. Elevated ALDH1B1 and NF- κ B activation in patient tumor tissues correlate with metastatic recurrence, underscoring their role in metastasis. While the association of ALDH1B1 upregulation with tumor metastasis and survival has been previously established, the authors have made significant strides in elucidating the mechanism by which tumors adapt to compressive forces within confined 3D spaces such as capillaries. With moderate revisions to the text and figures, this manuscript has the potential to be a strong fit for publication in a rigorous journal like Nature Communications.

Response: We thank the reviewer for the positive comments and constructive suggestions, which greatly strengthens the manuscript.

Major concerns:

Point 1. Authors should explain the screen setup in more details. For example, when dissecting mouse lung tissue, how to differentiate cancer cell in the alveoli or in the lung capillary tract? Also, authors need to show some basic quality control metric of the screen, at least in the supplementary to demonstrate the coverage and selection power of the screen.

Response: Thanks for your insightful comments.

1. According to reviewer's suggestion, we have supplemented more detailed descriptions of the screening criteria in methods section (Page: 31, line: 869) and figure legends (Extended Data Fig. 1a), and refined the screening flowchart (Fig. 1a) with additional details to clearly highlight the key steps in the screening process.

2. Over 90% of tumor cells remained lodged in the lung capillary tract at the 24 h post-tail vein injection in our mouse model, consistent with the previous studies that the majority of tumor cells reside in the pulmonary capillaries within 24 hours after tail vein injection (PMID: 10613833, PMID: 26855844).

3. The basic quality control data for CRISPR library: >95% of sgRNAs were effectively detected. In CRISPR screening, approximately 11.9% of sgRNAs were depleted in the control group (average reads/sgRNA = 273), while 19.5% were depleted in the experimental group (average reads/sgRNA = 231), indicating selection power of the screening system. As suggested by the reviewer, these data are presented in the Extended Data Fig. 1a.

Point 2. PIEZO1 and its sensing of the compression force were shown to affecting the

expression level of ALDH1B1. Does depletion of PIEZO1 therefore also affecting the survival of the tumor cells in tight capillaries? It seems this part of data is missing from the Figure 3.

Response: As suggested by the reviewer, we depleted PIEZO1 in tumor cells and observed that PIEZO1 depletion markedly increased tumor cell death in confining pores (Fig. 3k). Animal experiments showed that PIEZO1 depletion accelerated the diminishment of bioluminescence signals of tumor cells in the lungs (Fig. 3l). Further TUNEL assay on these lung tissues showed that PIEZO1 depletion increased tumor cell death in lung capillaries (Fig. 3m). Additionally, we found that PIEZO1 depletion inhibited ALDH1B1 expression in tumor cells confined within lung capillaries (Fig. 3n). These results were included in our revised manuscript (Page: 11, line: 290).

Point 3. Liu et al. have elucidated the signaling pathway from compression force sensing to upregulation of ALDH1B1. However, there remain one missing piece of the puzzle: how PIEZO1 activate CSK23? PIEZO1 is a calcium channel that activated by force, could there be a calcium sensing messenger between PIEZO1 and CSK23?

Response: Thank you for the insightful suggestion. The calcium signals are normally transduced by calcium-sensing messengers such as calmodulin (CaM), calpain, and calcineurin to regulate downstream signaling networks (PMID: 28622523). Among these, CaM emerges as a central Ca^{2+} sensor and molecular integrator, dynamically regulating protein-protein interactions and scaffolding macromolecular complex assembly through its target-binding domains (PMID: 2479144, PMID: 29247668). Thus, we tested whether CaM is the calcium messenger between PIEZO1 and CSK23. We treated A549 and H1299 cells with CaM-selective inhibitor W7 and observed that W7 treatment markedly attenuated compression-induced interaction between CSK23 and IKK β (Extended Data Fig. 9k,l). Meanwhile, we also assessed whether calpain or calcineurin also contributes to PIEZO1-mediated CSK23 activation. We treated A549 cells with the calpain inhibitor MDL-28170 or the calcineurin inhibitor FK506. Unlike W7, MDL-28170 treatment had no significant effect on the compression-induced CSK23-IKK β interaction, whereas FK506 treatment partially abrogated this interaction (Extended Data Fig. 9m,n). These results along with the results of Fig. 3 and Fig. 5 suggests that CaM is very likely the calcium messenger between PIEZO1 and CSK23. These results were included in our revised manuscript (Page: 13, line: 367).

Point 4. In Figure 6d, without any CSK23 depletion, IKKbeta pS177/181 level is lower in compression 1hr condition than the compression 0.5hr condition, which is contradicting with result in Figure 5b. What causes this discrepancy? The authors should quantify their western blot results and show statistical replicates.

Response: We apologized for the confusion. We reviewed the experimental records and concluded that this discrepancy likely resulted from using higher-passage A549 cells in original Fig. 6d. Thus, we revived low-passage A549 cells and repeated the experiments, which showed that IKK β pS177/181 level was higher at 1 hr than 0.5 hr upon compression (Fig. 6d), consistent with the result of Fig. 5b.

Minor concerns:

Point 1. The authors should introduce briefly (a sentence is sufficient) on the background of A549 cell line, which will help the reader understand rationale of using it.

Response: This point was well taken. We included a brief description of A549 cells (Page: 5, line: 90). “we conducted CRISPR knockout screening using lung adenocarcinoma-derived A549 cells-an epithelial cell line with robust migratory and invasive capacities.” (PMID: 9743595, PMID: 33659549).

Point 2. Figure 2a, right panel. It would be interesting to see the statistical comparison between group 1 and 3 (i.e., negative control vs shALDH1B1+ALDH1B1 rescue). It seems rescue on has even lower cell death rate.

Response: We reanalyzed the data and observed no significant difference between negative control group and shALDH1B1+ALDH1B1 rescue group.

Point 3. Please explain the assay briefly for Figure 2f. For example, what’s the green/red ratio. It would be easier to understand for the audience that are not familiar to ferroptosis.

Response: This point was well taken. We have expanded details in the Methods section (Page: 36, line: 1031). BODIPY 581/591 C11 is a ratiometric fluorescent probe that quantitatively measures lipid peroxidation during ferroptosis. In its reduced state, the probe emits red fluorescence (ex/em: 581/591 nm), which shifts to green fluorescence (ex/em: 488/510 nm) upon oxidation by lipid hydroperoxides. The relative lipid peroxidation (originally termed “green/red ratio”) was calculated as the intensity ratio of oxidized (green) to reduced (red) fluorescence signals (F510/F591). We have uniformly designated “green/red ratio” as “Rel. lipid peroxidation” in the current manuscript.

Point 4. Although treatment is not the focus of the study, the authors should attempt to discuss about the therapeutic potential (pros and cons) of ALDH1B1 inhibitor (or any method perturbing this cancer adaption pathway).

Response: As suggested by the reviewer, we added a discussion on the therapeutic promise and clinical challenges of ALDH1B1 inhibitor as well as CSK23 inhibitor (Page: 17, line: 475):

Our multimodal validation establishes ALDH1B1 as a therapeutically viable target for metastatic intervention. Both genetic depletion and pharmacological inhibition of ALDH1B1 significantly reduced metastatic burden and extended median survival in nude mouse models, highlighting its potential to suppress metastatic recurrence in post-operative cancer patients. However, clinical translation requires careful mitigation of on-target toxicities, primarily due to ALDH1B1's physiological roles in intestinal stem cell maintenance (PMID: 24789370) and pancreatic regeneration (PMID: 31548432). We propose targeting circulating tumor cells during early dissemination-a phase characterized by limited population size and unique metabolic dependencies, which may enable effective metastasis blockade via transient, submaximal ALDH1B1 inhibition. Additionally, ALDH1B1 inhibitors must address the

challenge of compensatory activation by other ALDH isoforms, a critical drawback of isoform-selective agents, to enhance therapeutic efficacy and minimize potential resistance mechanisms. Current broad-spectrum aldehyde dehydrogenase (ALDH) inhibitors, such as disulfiram (DSF) and diethylaminobenzaldehyde (DEAB), exhibit promising antitumor activity by simultaneously targeting multiple ALDH isoforms. Although their pan-ALDH inhibitory mechanism circumvents functional compensation, clinical advancement is still hampered by on-target toxicities stemming from broad-spectrum activity, including dose-limiting hepatotoxicity, cardiotoxicity, and off-tissue damage to vulnerable cell populations (PMID: 31515079).

Although CSK23, a member of the CK2 family, has been associated with lung cancer susceptibility (PMID: 20625391) and exhibits debated tumor-suppressive activity in hepatocellular carcinoma (PMID: 33582094), its broader role in cancer-particularly in metastasis-remains poorly characterized. In our study, we demonstrate the critical role of CSK23 in brain metastasis of lung cancer cells. Pharmacological inhibition by silmitasertib (CX-4945), a clinically available inhibitor targeting CSK23, significantly reduced metastatic burden and improved survival, suggesting that targeting CSK23 represents a promising strategy to prevent post-surgical metastatic recurrence in lung cancer.

Reviewer #2 (Remarks to the Author):

Reviewer Summary:

This manuscript presents a CRISPR knockout screen targeting 1,684 metabolic enzymes in a mouse model of lung cancer metastasis. The authors identify ALDH1B1, a member of the aldehyde dehydrogenase family, as essential for tumor cell survival within physically confining capillaries. Mechanistically, they propose that compressive forces activate casein kinase 2A3 (CSNK2A3), which in turn phosphorylates IKK β (I κ B kinase subunit beta) at serine residues 177 and 181. This phosphorylation event is suggested to activate the NF- κ B pathway, leading to upregulation of ALDH1B1 expression. The increased expression of ALDH1B1 is proposed to enhance aldehyde detoxification, suppressing ferroptosis and supporting tumor cell survival in confined environments, ultimately facilitating lung cancer metastasis. They further suggest that genetic or pharmacological inhibition of CSK23 or ALDH1B1 could impair lung cancer metastasis.

The central novelty and impact of the study lies in linking mechanical compression within capillaries to changes in ALDH isoform expression and promoting metastasis via ferroptosis suppression. The signaling work is of high quality and rigor, and the graphical summary at the end of the manuscript is especially well-done. However, despite the novelty, there are several significant concerns that must be addressed before publication:

Response: We thank the reviewer for the positive comments and insightful suggestions, which greatly strengthens the manuscript.

Point 1. These was limited scope of ALDH isoform screening. Specifically, the ALDH family includes 18 isoforms, many of which are implicated in cancer progression and metastasis. It is unclear whether all ALDH isoforms were included in the gRNA screen targeting 1,684 metabolic enzymes. Since isoform redundancy is well-documented—where suppression of one ALDH isoform results in compensatory upregulation of another—focusing solely on ALDH1B1 is insufficient. The manuscript must clarify whether other ALDH isoforms were also screened and ruled out. Controls targeting other ALDH isoforms (e.g., ALDH1A3, ALDH2, ALDH3A1) should be included to confirm the specificity of the phenotype.

Response: Thank you for the insightful suggestions.

1. Our CRISPR screening targeting 1684 metabolic enzymes comprehensively covered all 18 ALDH isoforms, including ALDH1A1, ALDH1A2, ALDH1A3, ALDH1B1, ALDH1L1, ALDH1L2, ALDH2, ALDH3A1, ALDH3A2, ALDH3B1, ALDH3B2, ALDH4A1, ALDH5A1, ALDH6A1, ALDH7A1, ALDH8A1, ALDH9A1, ALDH18A1.

2. To identify the metabolic enzyme required for tumor cell migration in confining spaces *in vivo*, we performed CRISPR screening, which identified ALDH1B1 as the most significantly enriched ALDH isoform, with ALDH6A1 and ALDH9A1 emerging as secondary candidates (Extended Data Fig. 2d). To determine the role of ALDH6A1 and ALDH9A1 on tumor cell survival in confining spaces, we established ALDH6A1 and ALDH9A1 depleted A549 cells (Extended Data Fig. 2e,f). ALDH9A1 depletion increased tumor cell death in confining pores (Extended Data Fig. 2g), though its effect was much weaker than ALDH1B1 ablation (Extended Data Fig. 2c). In contrast, ALDH6A1 depletion showed no significant effects on confined tumor

cell survival (Extended Data Fig. 2h). We also examined ALDH9A1 expression under mechanical compression. Quantitative PCR (qPCR) analysis revealed modest but statistically upregulation of ALDH9A1 under compression (Extended Data Fig. 6u,v). However, this mechanosensitive upregulation was markedly attenuated compared to ALDH1B1's response (Fig. 3g and Extended Data Fig. 6m). Transcriptomic analysis of the GSE288929 dataset (A549 cells) further demonstrated that baseline ALDH1B1 expression exceeded ALDH9A1 levels (Extended Data Fig. 2i), potentially explaining the dominant functional role of ALDH1B1 in confining spaces. Collectively, among ALDH isoforms, ALDH1B1 emerges as the master regulator governing tumor cell survival under spatial confinement. These results were included in the revised manuscript (Page: 6, line: 124 and page: 10, line: 277).

Point 2. There are limitations of the experimental metastasis assay that are not addressed. Specifically, in experimental metastasis assays, many cancer cells accumulate in the lungs initially, but the majority are naturally cleared within 24 hours. The authors need to clarify whether the genes identified are truly mediating survival in the confined capillaries or merely affecting initial accumulation. A clear distinction between transient lodging and sustained survival in capillaries must be made.

Response: Thank you for the constructive comment.

1. Initial lodging (30 min post-tail vein injection): No significant difference in fluorescence intensity was observed between ALDH1B1 depletion and control groups, indicating that ALDH1B1 depletion does not affect transient cell lodging (Fig. 1e, right, top lane).

2. Sustained survival (24 h post-tail vein injection): However, in sustained survival assays, ALDH1B1 depletion significantly accelerated the decline in the bioluminescence signals of tumor cells in lungs (Fig. 1e, right, bottom panel). Moreover, TUNEL assay on dissected lung tissues showed that ALDH1B1 depletion dramatically increased tumor cell death in lung capillaries (Fig. 1f). These results establish ALDH1B1 as a regulator of cell survival in confinement, rather than a mediator of transient lodging.

Point 3. Figures as presented suggest cell line limitation and lack of generalizability. Specifically, the study relies almost entirely on the A549 lung cancer cell line. To demonstrate robustness and reproducibility, results must be validated across additional lung cancer cell line and ideally across different cancer types. Partial validation of some experimentation in only the H1299 cell line is not sufficient. Figures throughout the manuscript should be updated to include these validations.

Response: This point was well taken.

1. The H1299 cells have been comprehensively validated for all critical experimental findings in original manuscript, including: (1) the role of ALDH1B1 in tumor cell survival in confining space (Extended Data Fig. 3 and Extended Data Fig. 5); (2) NF- κ B-mediated transcriptional regulation of ALDH1B1 in confining space (Extended Data Fig. 6, Extended Data Fig. 7 and Extended Data Fig. 8) and (3) the mechanotransduction cascade wherein confinement activates ALDH1B1 through the CSK23-IKK β -NF- κ B axis (Extended Data Fig. 9 and Extended Data Fig. 10).

2. As suggested by the reviewer, we expanded our validation to the H460 lung cancer cells.

1) We depleted ALDH1B1 in H460 cells and rescued the cells with wild-type ALDH1B1 (WT) or ALDH1B1 enzymatic-dead (ED) mutant (Extended Data Fig. 3d). We found that ALDH1B1 depletion markedly increased tumor cell death in confining pores (Extended Data Fig. 3f). Transwell migration assay showed that ALDH1B1 depletion greatly suppressed the confined migration of tumor cells (Extended Data Fig. 3h), while rescued expression of ALDH1B1 WT almost completely restored the survival and migratory capability of ALDH1B1-depleted cells in confining pores, excluding the off-target possibility of ALDH1B1 shRNA (Extended Data Fig. 3f,h). However, tumor cells rescued with ALDH1B1 ED could not recover the survival and the migratory capability of ALDH1B1-depleted cells in confinement.

2) We also explored whether compressive force induces ALDH1B1 upregulation in confined H460 cells. Our results showed that the mRNA and protein levels of ALDH1B1 were dramatically upregulated in the cells after compression in a time-dependent manner (Extended Data Fig. 6o,p).

3) We also established CSK23-depleted H460 cells (Extended Data Fig. 9d), and found that CSK23 depletion inhibited IKK β S177/181 and I κ B α S32/36 activation under compressive force (Extended Data Fig. 9g).

4) We also examined whether CSK23 regulates cell survival in confined cells by activating NF- κ B, we overexpressed RelA in CSK23-depleted H460 cells (Extended Data Fig. 10c). We found that CSK23 depletion increased cell death rate and inhibited the migratory capability of tumor cells in confining spaces, while RelA overexpression restored the survival and migration of CSK23-depleted cells in confinement (Extended Data Fig. 10h,k). Meanwhile, overexpression of ALDH1B1 can also restore the survival and migration of CSK23-depleted H460 cells in confining spaces (Extended Data Fig. 10n,q,s).

Point 4. There is concern related to the scope of confinement in the metastatic cascade. Specifically, the metastatic process involves several stages of physical confinement—not just within capillaries but also during tissue migration, intravasation, circulation, and extravasation. The manuscript currently focuses only on capillary confinement. The authors must experimentally address whether the described mechanism applies more broadly to other stages of the metastatic cascade.

Response: Thank you for the constructive suggestions.

As suggested by the reviewer, we also investigated whether ALDH1B1 regulates tumor cell survival during extravasation through endothelial gaps. Most surviving tumor cells begin extravasation between 24 and 36 hours after intravenous injection (PMID: 31243371, PMID: 20530574, PMID: 26855844). Since visualizing the survival of cells trapped within endothelial gaps proved challenging, we instead quantified both the cells remaining in capillaries and those that had completed extravasation 48 hours after injection, when most cells completed extravasation. Immunofluorescence analysis showed that ALDH1B1 depletion led to a 2-fold reduction in tumor cells retained within capillaries and a 5-fold decrease in extravasated tumor

cells (Fig. 1g), suggesting that ALDH1B1 likely also contributes to the survival of tumor cells during extravasation. These data are now included in the revised manuscript (Page: 6, line: 137).

Point 5. There is major concern regarding the possible redundancy among ALDH isoforms in metastasis. Specifically, given the overlap in detoxification function across ALDH isoforms, it's likely that multiple isoforms could suppress ferroptosis under confined conditions. The manuscript must explore and experimentally demonstrate whether other ALDH isoforms are also upregulated or functionally active in response to mechanical stress and assess their potential contributions.

Response: We thank the reviewer for their insightful suggestions.

1. Our CRISPR screening targeting 1684 metabolic enzymes comprehensively covered all 18 ALDH isoforms, including ALDH1A1, ALDH1A2, ALDH1A3, ALDH1B1, ALDH1L1, ALDH1L2, ALDH2, ALDH3A1, ALDH3A2, ALDH3B1, ALDH3B2, ALDH4A1, ALDH5A1, ALDH6A1, ALDH7A1, ALDH8A1, ALDH9A1, ALDH18A1.

2. To identify the metabolic enzyme required for tumor cell migration in confining spaces *in vivo*, we performed CRISPR screening, which identified ALDH1B1 as the most significantly enriched ALDH isoform, with ALDH6A1 and ALDH9A1 emerging as secondary candidates (Extended Data Fig. 2d). To determine the role of ALDH6A1 and ALDH9A1 on tumor cell survival in confining spaces, we established ALDH6A1 and ALDH9A1 depleted A549 cells (Extended Data Fig. 2e,f). ALDH9A1 depletion increased tumor cell death in confining pores (Extended Data Fig. 2g), though its effect was much weaker than ALDH1B1 ablation (Extended Data Fig. 2c). In contrast, ALDH6A1 depletion showed no significant effects on confined tumor cell survival (Extended Data Fig. 2h). We also examined ALDH9A1 expression under mechanical compression. Quantitative PCR (qPCR) analysis revealed modest but statistically upregulation of ALDH9A1 under compression (Extended Data Fig. 6u,v). However, this mechanosensitive upregulation was markedly attenuated compared to ALDH1B1's response (Fig. 3g and Extended Data Fig. 6m). Transcriptomic analysis of the GSE288929 dataset (A549 cells) further demonstrated that baseline ALDH1B1 expression exceeded ALDH9A1 levels (Extended Data Fig. 2i), potentially explaining the dominant functional role of ALDH1B1 in confining spaces. Collectively, among ALDH isoforms, ALDH1B1 emerges as the master regulator governing tumor cell survival under spatial confinement. These results were included in the revised manuscript (Page: 6, line: 124 and page: 10, line: 277).

Point 6. The therapeutic implications are overstated in several parts of the manuscript. Specifically, the suggestion that ALDH1B1 could be a promising therapeutic target is problematic. Many prior efforts to target specific ALDH isoforms have failed due to functional compensation by other isoforms. A more realistic approach would be to explore broad-spectrum ALDH inhibitors, which is not currently tested or discussed in the manuscript.

Response: We appreciate the reviewer's concerns on the challenges of ALDH isoform-specific targeting. A discussion on the therapeutic promise and clinical challenges of broad-spectrum ALDH inhibitors were included in the revised manuscript (Page: 17, line: 475):

While broad-spectrum ALDH inhibitors such as disulfiram (DSF) and

diethylaminobenzaldehyde (DEAB) can overcome functional compensation among isoforms, their clinical translation has been limited by on-target toxicities-including dose-limiting hepatotoxicity, cardiotoxicity, and off-tissue damage to vulnerable cell populations-due to indiscriminate inhibition of multiple ALDH family members (PMID: 31515079).

In contrast, our study provides multimodal evidence that ALDH1B1 may represent a more tractable target. For example, CRISPR/Cas9 screens specifically identified it as one of the most enriched essential genes for tumor cell survival under spatial confinement compared to other ALDH isoforms. Both genetic depletion and pharmacological inhibition of ALDH1B1 significantly reduced metastatic burden and extended median survival in preclinical models without observed compensatory mechanisms by other ALDH isoforms. These findings demonstrate ALDH1B1's critical role in sustaining tumor cell survival within capillary networks, while simultaneously revealing its therapeutic promise as a target for inhibiting metastatic dissemination.

Furthermore, a dual-inhibition strategy targeting both ALDH1B1 and ALDH9A1 may enhance therapeutic efficacy while demonstrating a more favorable toxicity profile compared to pan-ALDH inhibitors.

Point 7. Calibration of the in vitro model validation for mechanical compression is a concern. Specifically, the BioPress well system used to model mechanical compression must be better validated. The authors need to demonstrate that the in vitro forces and extracellular matrix (ECM) conditions are comparable to those experienced by tumor cells in capillaries or at other sites in the metastatic cascade. The contribution of ECM stiffness to gene expression changes should also be evaluated since it is significant in the scientific literature.

Response: Thank you for the constructive comments.

1. The BioPress well system has been extensively used in multiple peer-reviewed studies (e.g., PMID: 36444299), establishing it as a reliable platform for precise modeling of mechanical compression in cellular microenvironments.

2. The diameters of pulmonary capillaries range from 5 to 8 μm (PMID: 23606929). In microfluidic channels with analogous dimensions (5-8 μm), individual tumor cells experience compressive forces of approximately 80 nN (PMID: 33060332). Using the FlexCell system's standard conversion formula, $P \text{ (MPa)} = (5.65 \times \text{Force (lbs)}) / D \text{ (mm}^2\text{)}$, we determined that an applied pressure of 5 kPa in BioPress well system is equivalent to achieve the target cellular force load of 80 nN. We have included more detailed descriptions of the selection of in vitro applied forces in the Methods section (Page: 33, line 924).

3. To determine whether matrix stiffness regulates ALDH1B1 expression, we used a photo-crosslinked GelMA 3D hydrogel system replicating physiological stiffness gradients. Based on reported lung tissue mechanics (0.5-5 kPa; PMID: 29274784), we tested three GelMA concentrations representing distinct ECM stiffness regimes: 10% (~3 kPa, approximating normal lung), 12.5% (~5 kPa, mimicking fibrotic regions), and 15% (~9 kPa, representing pathological stiffness). Western blot analyses revealed no stiffness-dependent modulation of ALDH1B1 expression (Extended Data Fig. 6w), suggesting that ALDH1B1 regulation is independent of matrix stiffness. These results were included in the revised manuscript (Page: 10, line: 281).

Point 8. Western blot quality and quantification needs improvement. Specifically, several western blot images appear overexposed for the loading controls, which may not be within the linear detection range. The authors should provide lighter exposures and include quantitative analysis normalized to loading controls.

Response: This point was well taken. As suggested by the reviewer, we have now presented images with optimized exposure time (Fig. 3h, 3J, 4b, 5a, 5b, 5e, 5h, Extended Data Fig. 7b and Extended Data Fig. 8j). Additionally, we included quantitative analysis normalized to the loading control. (Fig. 3e, 3h, 4b, 4f, 6f, Extended Data Fig. 6p and Extended Data Fig. 9i).

Point 9. There is weakness of human tumor correlation data (Figure 7). Specifically, the correlative data from human tumors presented in Figure 7 are interesting but lack depth. Stronger validation using larger datasets or orthogonal approaches (e.g., IHC, RNA-seq, or survival analysis) would improve the translational relevance of this finding.

Response: Thank you for the constructive suggestion. Since conventional IHC and RNA-seq cannot reliably distinguish between intravascular tumor cells and extravascular tumor cells, we strengthened the clinical relevance of NF- κ B-dependent regulation of ALDH1B1 expression by expanding the cohort to include 61 primary lung cancer samples. Immunofluorescence staining and quantitative image analysis were performed on all tumor samples. The updated results are presented in Fig. 7. These results were included in the revised manuscript (Page: 15, line: 420).

Reviewer #3 (Remarks to the Author):

Reviewer Summary:

The paper from Liu et al (NCOMMS) proposes that compression force in capillaries activates NF- κ B during metastasis to promote survival of tumor cells through upregulation of ALDH1B1. The authors have used metastasis models (tail vein injection and intra-cardiac injection) to measure metastasis and analyze tumor cells in capillaries. In vitro studies, using confined pore transwell assays indicate that CSK23 is activated to phosphorylate IKK to promote NF- κ B activation which drives expression of ALDH1B1. Mutant ALDH1B1 is inactive in promoting the confinement-induced cell survival response, consistent with the effect of ALDH1B1 on acetaldehydes.

While the cell-based studies are consistent with the hypothesis, there is a concern with the tumor metastasis studies. Human lung cancer patient samples were analyzed to correlate with animal and cell-based studies. The authors have failed to acknowledge the extensive publication history on ALDH1B1 in cancer (over a 100 publications), where it is described to promote a variety of oncogenic phenotypes such as control of cancer stem cells and beta-catenin signaling (both involved in metastasis). Lack of consideration of these oncogenic phenotypes confounds interpretation of some of the proposed results. Additional concerns are outlined below.

Response: We thank the reviewer for the constructive comments and insightful suggestions, which greatly strengthens the manuscript. In our discussion, we have incorporated existing reports on ALDH1B1's functional roles in cancer and contextualized these findings alongside our own discoveries.

Point 1. In Fig. 1, the authors conclude that loss of ALDH1B1 in tumors promotes metastasis via effects on survival of tumor cells in capillaries. Given the significant role of ALDH1B1 in several cancer phenotypes (unreferenced, and see statement above), the authors cannot conclude that the loss on metastasis is solely through cell death in capillaries. In this regard, can the authors show that the same number of cells reach the lung for WT vs ALDH1B1 knockdown? Is that what is meant to be shown in fig. 1g? The authors need to analyze the number of mets that form and not total GFP intensity since loss of ALDH1B1 will likely reduce the growth of the metastases.

Response: Thank you for the constructive comments.

1. Accumulating evidence highlights that tumor cell survival in confining space represents a critical bottleneck step in metastatic progression (PMID: 27909339, PMID: 35256052, PMID: 37296072). Our study thus focused on the role of ALDH1B1 in tumor cell survival under spatially confining spaces. We agree with the reviewer that ALDH1B1 may also contribute to metastasis through other mechanisms, such as the regulation of cancer stem cell and β -catenin signaling. We thus modified the original statement to "These results demonstrate that ALDH1B1 is required for tumor metastasis at least partially by enhancing tumor cell survival within capillaries and during extravasation" (Page: 7, line: 165). Additionally, we have incorporated existing reports on the role of ALDH1B1 in metastasis and contextualized these findings alongside our own discoveries in the revised discussion (Page: 16, line: 458).

2. At 0.5 h post tail vein injection, bioluminescence imaging of the mice showed comparable

luciferase activity was observed between control cells and ALDH1B1-depleted cells (Fig. 1e), indicating a similar number of tumor cells reach the lung tissue. We have incorporated this result in the revised manuscript (Page: 6, line: 142).

3. As suggested by the reviewer, we re-analyzed the effects of ALDH1B1 depletion on the number of brain metastases, which showed that ALDH1B1 depletion significantly reduced the number and size of brain metastatic lesions (Fig. 1i). These results are described in the revised manuscript (Page: 7, line: 162).

Point 2. The authors utilize TCGA data to indicate that elevated levels of ALDH1B1 mRNA is linked with metastasis and survival. Related to the point above, the authors use Kaplan-Meier human data to argue that ALDH1B1 promotes metastasis. In the cases with higher ALDH1B1, it could be that the primary tumor is larger, has more cancer stem cells, or is resistant to therapy. Additionally, what is measured in the Kaplan-Meier analysis is ALDH1B1 RNA expressed in the tumor and not in tissue capillaries, thus making this point disconnected from the proposed mechanism of upregulation in confined capillaries. The lack of consideration of additional oncogenic phenotypes for ALDH1B1 is a significant flaw in the manuscript.

Response: Thank you for the constructive comments.

1. We have fully acknowledged that ALDH1B1 can contribute to tumor metastasis through multiple mechanisms. Our study uncovers a new mechanism of ALDH1B1-promoted tumor metastasis by supporting tumor cell survival under spatial confinement. In Discussion section (Page: 16, line: 458), we have systematically discussed multifaceted mechanisms orchestrated by ALDH1B1 in tumor metastasis (e.g., cancer stemness maintenance, therapy resistance).

2. We agree with the reviewer that the correlation of ALDH1B1 expression with metastasis and survival in TCGA data may be due to its possible associations with primary tumor growth, cancer stem cell expansion, and therapy resistance. However, this TCGA data analysis, conducted immediately after the screening (Fig. 1a-1c), mainly serves as a crucial initial filter to exclude metabolic enzymes less relevant to metastasis, as identified in the CRISPR screen.

3. We acknowledge the reviewer's point that, due to the limitations of bulk sequencing data from TCGA, the Kaplan-Meier analysis reflects ALDH1B1 RNA expression in tumor tissue rather than specifically in capillaries. To address this limitation, we performed immunofluorescence staining of ALDH1B1 in lung cancer patient specimens, which demonstrated that elevated ALDH1B1 expression specifically in intravascular tumor cells strongly correlates with metastatic recurrence (Fig. 7).

Point 3. For the studies shown in Fig. 2 related to survival and other effects on confined transwell pore assays, the authors need to show side by side analysis with unconfined cells. In the cell studies related to suppression of ferroptosis by ALDH1B1, did the authors analyze apoptosis? (this relates to a point below).

Response: Thank you for the constructive comments.

1. As suggested by the reviewer, we reinstated the data from unconfined cells, which showed that ALDH1B1 depletion did not significantly influence lipid peroxidation (Extended Data Fig. 5a,b) or survival (Fig.2a and Extended Data Fig. 3e,f) in unconfined tumor cells. These results

are described in the revised manuscript (Page: 6, line: 122 and page: 8, line: 205).

2. We really appreciate this constructive comment from the reviewer. As suggested by the reviewer, we determined whether ALDH1B1 regulates cell apoptosis under confinement by examining caspase-3/7 activity in ALDH1B1-depleted tumor cells, which showed that caspase-3/7 activity was elevated in confined cells, which was exacerbated by ALDH1B1 depletion (Extended Data Fig. 5i). However, compared to its effect on apoptosis, ALDH1B1 depletion exerts a significantly stronger impact on ferroptosis in confined tumor cells (Extended Data Fig. 5a,b). These results are consistent with previous studies demonstrating that aldehydes induce both apoptosis (PMID: 16040627) and ferroptosis (PMID: 36274088), and that aldehydes derived from ALDH inhibition tend to promote ferroptosis (PMID: 37552043, PMID: 40233740, PMID: 32458004). Collectively, these results suggest that ALDH1B1 plays a dual protective role in confined tumor cells by suppressing both ferroptosis and apoptosis, with its anti-ferroptotic effect being mechanistically dominant. These results are described in the revised manuscript (Page: 9, line: 222).

Point 4. For the metastasis studies using the ALDH1B1 inhibitor, again the authors need to show effects on metastasis numbers and size, and not luciferase intensity (this is focused on the concept that inhibiting ALDH1b1 will likely block growth of mets – unrelated to effects on actually promoting metastasis).

Response: Thank you for the insightful suggestion. We re-analyzed the effects of ALDH1B1 inhibitor treatment on the number of brain metastases and found that pharmacological inhibition of ALDH1B1 significantly reduced the number and size of brain metastatic lesions. These results are described in the revised manuscript (Page: 9, line: 242).

Point 5. Fig. 3 shows that the mechanosensory protein PIEZO1 is required for activation of ALDH1B1. Effects are called “dramatic” but are 2-fold or less.

Response: We have modified the description to remove “dramatically” in the manuscript to more precisely reflect the quantitative changes (Page: 10, line: 264).

Point 6. PDTC blocked the upregulation of ALDH1B1, and the authors took to mean that the IKK/NF- κ B pathway was involved. Note that PDTC can block a number of pathways. They authors note that levels of RelA are increased in compression/confinement but this experiment does not measure nuclear RelA. Depletion of RelA blocked compression/confinement-induced ALDH1B1. The authors linked confinement induced cell death with ferroptosis (and see comment below). As before, confinement results with non-confinement results need to be placed together.

Response: Thank you for the insightful comments.

1. We appreciate the reviewer’s note regarding PDTC, a known inhibitor of NF- κ B signaling (PMID: 28712932, PMID: 10491379, PMID: 2103049). Although PDTC may have off-target effects, our subsequent experiments using RelA-depleted cells confirmed the NF- κ B pathway’s involvement. Specifically, RelA depletion significantly downregulated ALDH1B1 expression

in confined tumor cells (Fig. 4f and Extended Data Fig. 7f).

2. In addition, as shown in Fig. 5c, compression induced a clear increase in nuclear RelA levels.
3. As suggested, we reinstated the data from unconfined RelA-depleted cells, which showed that RelA depletion did not significantly affect lipid peroxidation (Fig. 4k and Extended Data Fig. 7j) or survival (Fig. 4l and Extended Data Fig. 7k) in unconfined tumor cells. These results are described in the revised manuscript (Page: 12, line: 328).

Point 7. Compression induces the phosphorylation of IKKbeta, and this correlated with p-IkBa (in the figures the authors need to refer to this correctly – IkappaB-alpha - with Greek letters). In Fig 5d, the effect on an NF-κB reporter is relatively minimal.

Response:

1. Thanks for pointing it out. The corrections have been made in the revised manuscript.
2. Given that RelA nuclear translocation was enhanced as early as 0.5 hours post-compression (Fig. 5c), we subsequently evaluated NF-κB activity at the earlier 2-hour time point (compared to our previous 6-hour assessment). Our results demonstrated significantly elevated NF-κB activity at this earlier 2-hour time point. These results are included in the revised manuscript (Page: 12, line: 342).

Point 8. The authors found that CSK3 associates with IKKbeta during tumor cell compression and promotes IKKbeta activity, Knockdown of CSK3 leads to loss of ALDH1B1 expression and promoted cell death.

Response: We are a bit confused about the question. We speculate that the reviewer maybe want us to include the data of unconfined CSK23-depleted cells. Accordingly, we have reinstated the data from unconfined CSK23-depleted cells, which showed that CSK23 depletion did not significantly affect lipid peroxidation levels (Fig. 6h and Extended Data Fig. 10d, e and o) or survival (Fig. 6i and Extended Data Fig. 10f-h, p and q) in unconfined tumor cells. These results are described in the revised manuscript (Page: 14, line: 393).

Point 9. In Fig. 6, does knockdown of CSK3 and the CSK3 inhibitor block mets (number and size) – see points above. Like ALDH1B1, CSK3 is linked with oncogenic phenotypes – not referenced. The question is whether loss of CSK3 truly affecting metastasis or simply growth of the tumors once they have metastasized.

Response: Thank you for the insightful suggestion.

1. We re-analyzed the effects of CSK23 depletion and CSK23 inhibitor treatment on the number of brain metastases. The results demonstrate that both CSK23 depletion and pharmacological inhibition of CSK23 significantly reduced the number and size of brain metastatic lesions. These results are described in the in the revised manuscript (Page: 14, line: 398; Page: 14, line: 414).

2. We also included some discussion about other oncogenic functions of CSK23 (Page: 17, line: 496):

Although CSK23, a member of the CK2 family, has been associated with lung cancer

susceptibility (PMID: 20625391) and exhibits debated tumor-suppressive activity in hepatocellular carcinoma (PMID: 33582094), its broader role in cancer-particularly in metastasis-remains poorly characterized. In our study, we demonstrate the critical role of CSK23 in brain metastasis of lung cancer cells. Pharmacological inhibition by silmitasertib (CX-4945), a clinically available inhibitor targeting CSK23, significantly reduced metastatic burden and improved survival, suggesting that targeting CSK23 represents a promising strategy to prevent post-surgical metastatic recurrence in lung cancer.

Point 10. Fig. 7 links ALDH1 levels with capillary tumor cells and metastasis in lung cancer patients. These experiments are examining migration of cells from the tumor site to relate to distal metastasis and survival. There are several questions/concerns. When the authors refer to unconstrained tumor cells, are they simply referring to tumor? The cell death measurement is TUNEL assays which is associated with apoptosis and less with ferroptosis (see point above – did they authors measure apoptosis in their cell-based studies? Lastly, the text says that the authors are measuring nuclear RelA, but there is nothing to describe how that was measured. It seems that they are staining for whole cell RelA – but only nuclear would be meaningful for the overall hypothesis.

Response: Thanks for the constructive comments.

1. In this study, “unconstrained tumor cells” specifically refer to tumor cells located outside CD31-positive regions, indicating tumor cells that are not confined within capillaries. A more detailed description of this analytical process has been provided in the Methods section (Page: 34, line: 950).
2. We agree with the reviewer that TUNEL staining is more closely associated with apoptosis. In Extended Data Fig. 5i, we determined whether ALDH1B1 regulates cell apoptosis under confinement by examining caspase-3/7 activity in ALDH1B1-depleted tumor cells, which showed that caspase-3/7 activity was elevated in confined cells, which was exacerbated by ALDH1B1 depletion. However, compared to its effect on apoptosis, ALDH1B1 depletion has a much stronger influence on the ferroptosis of confined tumor cells (Extended Data Fig. 5a,b). These results are consistent with previous studies demonstrating that aldehydes induce both apoptosis (PMID: 16040627) and ferroptosis (PMID: 36274088), and that aldehydes derived from ALDH inhibition tend to promote ferroptosis (PMID: 37552043, PMID: 40233740, PMID: 32458004). Collectively, these results suggest that ALDH1B1 plays a dual protective role in confined tumor cells by suppressing both ferroptosis and apoptosis, with its anti-ferroptotic effect being mechanistically dominant. To strengthen the clinical relevance of ALDH1B1 expression to ferroptosis, we analyzed specific markers of ferroptosis. Since 4-hydroxynonenal (4-HNE) levels directly reflect intracellular aldehyde accumulation and lipid peroxidation (PMID: 28456642). Multiplex immunofluorescence analysis of primary lung cancer lesions revealed significantly higher 4-HNE expression in intravascular tumor cells (CD31⁺ regions) compared to extravascular areas (Fig. 7f). Moreover, ALDH1B1 expression exhibited a significant negative correlation with 4-HNE levels (Fig. 7g). Notably, tumor cells confined within capillaries from patients with metastatic recurrence displayed much lower 4-HNE levels than those from patients without metastatic recurrence (Fig. 7k). These results were included in the revised manuscript (Page: 15, line: 426).

3. Although we stained for total cellular RelA (including both cytoplasmic and nuclear localization), our quantitative analysis focused specifically on nuclear RelA intensity. Nuclear regions were defined based on DAPI staining, and RelA intensity was measured exclusively within these demarcated areas. As suggested, we have now provided a more detailed description of this methodology in the Methods section (Page: 34, line: 971).

Reviewer #4 (Remarks to the Author)

Reviewer Summary:

The manuscript by Liu et al. investigates the role of ALDH1B1 in promoting lung cancer metastasis by enhancing cell survival and motility. The authors suggest that ALDH1B1 is regulated by the CSK23-RelA/NF- κ B pathway and propose Piezo1 as a mechanosensor that initiates this signaling pathway in lung cancer. The use of a CRISPR knockout screening system to identify key metabolic enzymes is a sophisticated and logically sound approach. However, the method used to identify Piezo1, specifically through the application of a tumor microenvironment-mimicking platform, is not appropriate for lung cancer. Given the presence of numerous other mechanosensitive calcium channels, such as those in the Orai or TRPV families, the authors have not sufficiently explored alternative possibilities for mechanosensing in lung cancer metastasis. Consequently, the claim that Piezo1 is the primary mechanosensor driving CSK23-RelA signaling lacks a strong logical foundation. Additionally, it remains unclear whether aldehyde levels are elevated in the confined space of the lung. The authors should demonstrate a clear relationship between aldehyde production and the metastatic progression of lung cancer to support their hypothesis. Furthermore, the study reports brain metastasis occurring within seven days after lung cancer inoculation into the lung. Is there any existing literature supporting such a rapid metastatic timeline? The authors should provide references or additional experimental validation for this observation. The definition and generation of confined space in this study also require further clarification. While lung cancer cells may experience mechanical constraints within capillaries during metastasis, anoikis or pulsatile hydrostatic pressure could play a more significant role than simple confinement. The authors have not considered the possibility that different types of mechanical stress might influence lung cancer behavior. A more comprehensive discussion of these factors is necessary to strengthen the study's conclusions.

Overall, while the study presents an interesting perspective on ALDH1B1-mediated metastasis, the manuscript requires a more logically structured approach, further experimental validation, and consideration of alternative mechanosensors and mechanical stress factors.

Response: We thank the reviewer for the constructive comments and insightful suggestions, which greatly strengthens the manuscript.

Point 1. The use of a CRISPR knockout screening system to identify key metabolic enzymes is a sophisticated and logically sound approach. However, the method used to identify Piezo1, specifically through the application of a tumor microenvironment-mimicking platform, is not appropriate for lung cancer. Given the presence of numerous other mechanosensitive calcium channels, such as those in the Orai or TRPV families, the authors have not sufficiently explored alternative possibilities for mechanosensing in lung cancer metastasis. Consequently, the claim that Piezo1 is the primary mechanosensor driving CSK23-RelA signaling lacks a strong logical foundation.

Response: Thank you for the insightful suggestions.

Multiple mechanosensitive calcium channels, including ORAI (PMID: 38672434) and TRPV family members (PMID: 15922584), play important roles in cancer progression (PMID:

37762011, PMID: 30086406). In response to the reviewer's comment, we have evaluated their contributions to confinement-induced ALDH1B1 upregulation. We generated two A549 cell lines with depletion of key ORAI1 (PMID: 22108917, PMID: 30414508) or TRPV4 (PMID: 21204499) subtypes (Extended Data Fig. 6g,h), we found that ORAI1 depletion had no significant effect on ALDH1B1 expression in confined tumor cells (Extended Data Fig. 6i,j). TRPV4 depletion partially abrogated ALDH1B1 upregulation (Extended Data Fig. 6k,l), though unlike PIEZO1 depletion which completely abrogated ALDH1B1 upregulation in confined tumor cells (Fig. 3d,e and Extended Data Fig. 6e,f). These results suggest that under spatial confinement, PIEZO1 may serve as the primary mediator of ALDH1B1 expression regulation. These results have been incorporated into the revised manuscript (Page: 9, line: 250).

Besides the tumor microenvironment (TME)-mimicking platform, we further confirmed the role of PIEZO1 in lung cancer metastasis with animal experiments, which showed that PIEZO1 depletion accelerated the diminishment of bioluminescence signals of tumor cells in the lungs (Fig. 3l). TUNEL assay with these lung tissues showed that PIEZO1 depletion increased tumor cell death in lung capillaries (Fig. 3m). Additionally, PIEZO1 depletion inhibited ALDH1B1 expression in tumor cells confined within lung capillaries (Fig. 3n). These results have been incorporated into the revised manuscript (Page: 11, line: 291).

Point 2. Additionally, it remains unclear whether aldehyde levels are elevated in the confined space of the lung. The authors should demonstrate a clear relationship between aldehyde production and the metastatic progression of lung cancer to support their hypothesis.

Response: Thank you for the insightful suggestion. As suggested, we investigated whether aldehyde levels are elevated in the confined lung microenvironment. 4-hydroxynonenal (4-HNE) is a well-established marker of aldehyde metabolism (PMID: 12893006) and a known substrate of ALDH1B1 (PMID: 20616185). Given that 4-HNE levels directly reflect intracellular aldehyde accumulation and lipid peroxidation (PMID: 28456642), we used 4-HNE as a surrogate measure for aldehyde production in spatially confined tumor cells. Immunofluorescence analysis of primary lung cancer lesions revealed significantly higher 4-HNE levels in intravascular tumor cells (CD31⁺ regions) compared to extravascular areas (Fig. 7f). Moreover, ALDH1B1 expression exhibited a significant negative correlation with 4-HNE levels (Fig. 7g). Notably, tumor cells confined within capillaries from patients with metastatic recurrence displayed much lower 4-HNE levels than those from patients without metastatic recurrence (Fig. 7k). Furthermore, we found that ALDH1B1 depletion increased 4-HNE accumulation in tumor cells confined in lung capillaries (Fig. 2j). These results demonstrate that aldehyde levels are elevated under confinement, while ALDH1B1 protects tumor cells by detoxifying aldehydes, thereby promoting metastatic progression. This result was included in our revised manuscript (Page: 8, line: 217 and page: 15, line: 426).

Point 3. Furthermore, the study reports brain metastasis occurring within seven days after lung cancer inoculation into the lung. Is there any existing literature supporting such a rapid metastatic timeline? The authors should provide references or additional experimental validation for this observation.

Response: We apologized for the confusion. In the study, we employed an intracardiac injection metastasis model, where A549 cells were directly introduced into the left ventricle to facilitate systemic circulation and subsequent brain metastasis formation. This established methodology is specifically designed to study late-stage hematogenous metastasis, particularly brain metastasis (PMID: 39037653). In this model, tumor cells reach the brain within 30 minutes and undergo gradual cell death. By day 63 post-injection, we could observe the formation of significant brain metastatic lesions. Based on our observation and published literature using this model (PMID: 39708324), the signals detected on day 7 were more likely from residual tumor cells rather than established metastatic lesions. The data from day 7 was used to support that ALDH1B1 depletion accelerates tumor cell death in brain. Detailed description of this mouse model of brain metastasis was provided in the Methods section (Page: 35, line: 1008).

Point 4. The definition and generation of confined space in this study also require further clarification. While lung cancer cells may experience mechanical constraints within capillaries during metastasis, anoikis or pulsatile hydrostatic pressure could play a more significant role than simple confinement. The authors have not considered the possibility that different types of mechanical stress might influence lung cancer behavior. A more comprehensive discussion of these factors is necessary to strengthen the study's conclusions.

Response: Thank you for the insightful suggestions. A more comprehensive discussion of various mechanical stresses that tumor cells may experience within capillaries was included in the revised discussion section (Page: 18, line: 518):

During metastatic dissemination, tumor cells encounter diverse mechanical stresses in distinct vascular environments. For example, anoikis primarily occurs in large vessels; pulsatile hydrodynamic pressure dominates in arteries; and in narrow capillaries, tumor cells mainly experience both fluid shear stress and mechanical compression (PMID: 28898694, PMID: 39917412, PMID: 11136289). Our study mainly focuses on mechanical compression, demonstrating that the CSK23-IKK β -ALDH1B1 pathway is critical for tumor cell adaptation to compressive forces, enabling their survival in constricted capillaries. Targeting this pathway impairs mechanical stress adaptation and suppresses distant metastasis.

In contrast, fluid shear stress has been extensively studied for its role in tumor metastasis, as it not only disrupts circulating tumor cells (CTCs) from settling but may also induce cell-cycle arrest or even cell death. To overcome this mechanical barrier, CTCs enhance their phenotypic plasticity (e.g., through EMT or cytoskeletal remodeling) to improve survival (PMID: 18310319, PMID: 28054593, PMID: 33238151). Additionally, they form aggregates with platelets, neutrophils, and other cells via adhesive interactions, which protect their surface from shear forces and NK-cell-mediated lysis (PMID: 24862136). Meanwhile, CTCs upregulate bulky glycoproteins on their surface to increase physical adhesion to endothelial walls, thereby resisting shear forces. These adaptive mechanisms enable tumor cells to overcome shear stress and successfully establish distant metastases (PMID: 25030168). However, whether the CSK23-IKK β -ALDH1B1 pathway is also critical for tumor cell adaptation to fluid shear stress within capillaries requires further investigation.

Reviewer #5 (Remarks to the Author):

Reviewer Summary:

In this manuscript, Liu et al. describe metabolic adaptations that enable cancer cells to survive and metastasize within confining microenvironment. Using an in vivo CRISPR-based knockout screening system, the authors identify ALDH1B1 as a critical target for tumor cell survival within capillaries. Mechanistically, they demonstrate that compressive forces activate CSK23, which phosphorylates IKK β at S177/181, triggering NF- κ B-dependent upregulation of ALDH1B1. Elevated ALDH1B1 enhances aldehyde detoxification, suppresses ferroptosis, and promotes tumor cell survival, highlighting CSK23/ALDH1B1 as potential therapeutic targets for metastatic cancer. Overall, this work uncovers a connection between mechanical stress, NF- κ B signaling, and aldehyde metabolism during metastatic progression. Addressing the major concerns would strengthen the manuscript, the detail review is outlined below:

Response: We thank the reviewer for the constructive comments and insightful suggestions, which greatly strengthens the manuscript.

Major points:

Point 1. The authors used tail vein injection of gene-edited A549 cells to observe genetic differences in cells within pulmonary capillaries. How did the authors exclude the possibility that gene knockout affected cell proliferation, leading to false-positive results of reduced vascular cell counts? Additionally, is this model original? Is there any other literature that uses this method?

Response: Thanks for the comments.

1. In the animal experiment (Fig. 1a), observations and sample collections were performed at 24 h. In circulation, most of the cells do not proliferate at 24 h. Moreover, as shown in Extended Data Fig. 4c and d, ALDH1B1 depletion did not significantly influence the proliferation of A549 cells.

2. The method of the visualization of tumor cell viability in lung tissues in tumor metastasis mouse model via tail vein injection was extensively used in many previous studies (PMID: 33993094, PMID: 30926774, PMID: 37296072).

Point 2. The authors proposed that confined cells sense mechanical stress through Piezo1 and ultimately express ALDH1B1 to mitigate aldehyde toxicity. Why is this pathway specific to confined cancer cells? Why do normally growing cells not require similar mechanisms to maintain growth? Is it because confined cells face higher ferroptotic stress? Please provide experimental data to clarify.

Response: Thank you for the question. Indeed confinement significantly increased lipid peroxidation in tumor cells (Extended Data Fig. 5a,b) and elevated cell mortality (Fig. 2a, and Extended Data Fig. 3e,f), consistent with the upregulated aldehyde levels in confined tumor cells (Fig. 2c and Extended Data Fig. 3i). These results have been incorporated into the revised manuscript (Page: 6, line: 122 and page: 8, line: 205).

Point 3. In Patient 2's results (not enlarged), ALDH1B1 expression showed poor co-expression with CD31, and high ALDH1B1 expression was apparently observed in non-confined regions. However, the statistical plot in Fig. 7b does not reflect this trend. The authors should check the representative results, carefully verify the statistical methods, and accurately present the statistical outcomes.

Response: Thank you for the comment. Upon re-examining the representative results from Patient 2, we confirmed that ALDH1B1-high regions predominantly co-localized with CD31-positive areas. The poor co-localization visibility in the merged image was due to the much stronger ALDH1B1 and RelA signals than CD31 signals in some regions. To avoid the confusion, we have now replace it with more clearly distinguishable representative results (Fig. 7a).

Minor comments:

Point 1. Western blot images in high-quality journals should include molecular weight markers.

Response: As suggested by the reviewers, we have now added molecular weight markers to all western blot results.

Point 2. No information on tumor patient tissues or ethical approval is provided in the manuscript.

Response: This point was well taken. Detailed clinical information of the cancer patients along with the ethical approval number (KS(Y)23082) and (XHEC-C-2025-155-1) have been documented in the Methods section (Page: 34, line: 972).

Point 3. Please confirm the appropriateness of using the GEO database (GSE72094), which includes tumors with Kras and other genes mutations.

Response: As suggested by the reviewers, we have confirmed that the GSE72094 dataset contains lung adenocarcinoma samples with KRAS and other gene mutations.

Point 4. In Fig. 1h, TUNEL-positive areas do not fully overlap with ALDH1B1-knockdown cells, and non-tumor dead cells may affect the result. The authors should re-select representative images or quantify the percentage of co-expression between TUNEL-positive and mCherry-positive regions.

Response: Thank you for the comments. Our quantification specifically measured the proportion of mCherry⁺/TUNEL⁺ tumor cells relative to total mCherry⁺ tumor cells. To avoid confusion, we have added new representative images and modified the axis label in Fig. 1f to "mCherry⁺ tumor cell death (%)".

Point 5. The method section should highlight the sampling of pulmonary capillaries.

Response: As suggested by the reviewers, we have added more detailed descriptions of the tumor cell sampling process in the Methods section (Page: 31, line: 869). Our sampling encompassed tumor cells from the entire lung, with over 90% of tumor cells remained lodged in the lung capillary tract at the 24 h post-tail vein injection in our mouse model, consistent with previous studies (PMID: 10613833, PMID: 26855844). We also updated the screening workflow diagram (Figure 1a) to improve understanding of our approach.

Point 6. Please carefully verify all results for data integrity (e.g., no endpoint data for the experimental group in Fig. 2m).

Response: In original Fig. 2m, the observation period was terminated upon the death of the last control group mouse. At this endpoint, the surviving mice in the experimental group remained in good condition, with no significant weight loss or other adverse effects observed. These surviving mice were right-censored to ensure both ethical compliance and statistical validity, as the Kaplan-Meier method inherently accounts for censored data (PMID: 30015653, PMID: 27713848, PMID: 21367692). We have now added a detailed description of this approach in the Methods section (Page: 36, line: 1018).

Point 7. Nuclear and cytoplasmic fractions in Fig. 5c should be presented on the same gel to compare.

Response: Thank you for the suggestion. The nuclear and cytoplasmic fractions were loaded and imaged on the same gel. However, because the cytoplasmic fraction had a significantly higher protein concentration (leading to overexposed p65 signals compared to the nuclear fraction), we used different exposure times for the cytoplasmic and nuclear fractions.

Point 8. Total protein loading controls are required for phosphoprotein detection in Fig. 6d.

Response: Thank you for the suggestions. As suggested by the reviewer, we have added actin as the reference.

Point 9. Supplementary images are excessively compressed, compromising readability.

Response: As suggested by the reviewer, we have revised the supplementary images to improve readability.

Reviewer #6 (Remarks to the Author):

Response: Thank you for your help.

Reviewer #7 (Remarks to the Author):

Response: Thank you for your help.

REVIEWER COMMENTS

Reviewer #6 (Remarks to the Author):

Reviewer Summary:

The authors have undertaken substantial revisions to the manuscript and have addressed most of my previous concerns. I appreciate their efforts and am generally pleased with the progress. Nonetheless, there remain a few critical issues that need further attention before the manuscript can be considered for publication.

Response: We thank the reviewer for the positive comments and constructive suggestions, which greatly strengthens the manuscript.

Minor concerns:

Point 1. In their response to Major Point 4 raised by Reviewer 1, the authors attribute the observed discrepancies to the use of higher-passage cells. While this explanation is plausible, it also demonstrates the significant experimental variation across batches. Without additional evidence, it is difficult to determine whether the observed effects are due to cell passage or are confounded by batch-specific factors. To strengthen the validity of their conclusion, I strongly recommend that the authors include biological replicates of the relevant experiments and provide appropriate statistical analyses. This will allow for a more rigorous assessment of reproducibility and help clarify the source of the observed variation.

Response: As suggested by the reviewer, we quantified the ratio of phospho-IKK β S177/181 to total IKK β expression of Fig. 6d from three biologically independent experiments. Quantitative analysis and statistical comparison (Fig. 6d, right) showed that higher phospho-IKK β S177/181 levels at 1 hr compared to 0.5 hr in the cells upon compression are reproducible and statistically significant.

Point 2. The identification of Calmodulin as a potential mediator in the signal transduction pathway between PIEZO1 and CSK23 is a noteworthy and potentially impactful finding. If the authors are confident in the robustness of this result, I suggest that they incorporate this finding into the graphical abstract presented in Figure 8.

Response: As suggested by the reviewer, we have updated Figure 8 to include Calmodulin (CaM) as a mediator in the signaling pathway between PIEZO1 and CSK23.

Review comments on the revised manuscript:

The authors have undertaken substantial revisions to the manuscript and have addressed most of my previous concerns. I appreciate their efforts and am generally pleased with the progress. Nonetheless, there remain a few critical issues that need further attention before the manuscript can be considered for publication:

1. In their response to Major Point 4 raised by Reviewer 1, the authors attribute the observed discrepancies to the use of higher-passage cells. While this explanation is plausible, it also demonstrates the significant experimental variation across batches. Without additional evidence, it is difficult to determine whether the observed effects are due to cell passage or are confounded by batch-specific factors. To strengthen the validity of their conclusion, I strongly recommend that the authors include biological replicates of the relevant experiments and provide appropriate statistical analyses. This will allow for a more rigorous assessment of reproducibility and help clarify the source of the observed variation.
2. The identification of Calmodulin as a potential mediator in the signal transduction pathway between PIEZO1 and CSK23 is a noteworthy and potentially impactful finding. If the authors are confident in the robustness of this result, I suggest that they incorporate this finding into the graphical abstract presented in Figure 8.